# BEAR reveals that increased fidelity variants can successfully reduce the mismatch tolerance of adenine but not cytosine base editors

András Tálas [1,2✉], Dorottya A. Simon[1,3], Péter I. Kulcsár [1,4,5], Éva Varga[1,6], Sarah L. Krausz[1,2] & Ervin Welker [1,7✉]

Adenine and cytosine base editors (ABE, CBE) allow for precision genome engineering. Here, Base Editor Activity Reporter (BEAR), a plasmid-based fluorescent tool is introduced, which can be applied to report on ABE and CBE editing in a virtually unrestricted sequence context or to label base edited cells for enrichment. Using BEAR-enrichment, we increase the yield of base editing performed by nuclease inactive base editors to the level of the nickase versions while maintaining significantly lower indel background. Furthermore, by exploiting the semi-high-throughput potential of BEAR, we examine whether increased fidelity SpCas9 variants can be used to decrease SpCas9-dependent off-target effects of ABE and CBE. Comparing them on the same target sets reveals that CBE remains active on sequences, where increased fidelity mutations and/or mismatches decrease the activity of ABE. Our results suggest that the deaminase domain of ABE is less effective to act on rather transiently separated target DNA strands, than that of CBE explaining its lower mismatch tolerance.

[1] Institute of Enzymology, Research Centre for Natural Sciences, Budapest, Hungary. [2] School of Ph.D. Studies, Semmelweis University, Budapest, Hungary. [3] Gene Design Ltd., Szeged, Hungary. [4] Department of Biophysics and Radiation Biology, Semmelweis University, Budapest, Hungary. [5] Biospiral-2006 Ltd., Szeged, Hungary. [6] School of Ph.D. Studies, University of Szeged, Szeged, Hungary. [7] Institute of Biochemistry, Biological Research Centre, Szeged, Hungary. ✉email: talas.andras@ttk.hu; welker.ervin@ttk.hu

Cas9 nucleases recognise DNA sequences that are located immediately upstream of their respective protospacer adjacent motif (PAM) sequences and are complementary to the ~20 nucleotide-long 5' part (spacer) of their single guide RNAs (sgRNAs)[1–4]. These nucleases facilitate effective genome modifications by introducing site specific double-strand breaks into the DNA; however, unwanted insertions or deletions (indels) also frequently fleck the modified genome[5,6]. In contrast, base editors have been designed to perform genome modifications without introducing DNA double-strand breaks. They generally comprise an RNA guided nuclease (Cas9 or Cas12a) fused to a deaminase enzyme, and together they are capable of inducing transition mutations. In theory, these mutations allow the correction of the majority of pathogenic single nucleotide variations found in the human genome[7–9]. Cytosine base editors (nCBEs or dCBEs, referring to the nickase or the inactive [dead] nuclease version, respectively) contain a cytidine deaminase that converts cytosines of the non-targeted DNA strand into uracils, which will then be replaced by thymines during DNA replication[7]. Similarly, adenine base editors (nABEs and dABEs) contain an adenosine deaminase that converts adenines of the non-targeted DNA strand into inosines which are read as guanines and therefore replaced during DNA replication[8].

Unfortunately, genome modification efficiency of nuclease inactive base editors, dCBEs and dABEs, lags behind that of SpCas9 nuclease and could be increased only by employing a nickase Cas9[7]. By nicking the non-modified DNA strand, the DNA repair systems are biased towards using the uridine- or inosine-containing strand as a template, which significantly improves the efficiency of base editing[7,8]. However, repairing the nicks generated by nickases yields varying amounts of unwanted indels in a target- and cell type-dependent manner, they have been reported to be as high as 10–20% in some cases[7,10,11]. One solution for increasing the yield of base editing without the generation of nick-induced indels would be to enrich cells that contain dABE- and dCBE-edited bases, using a marker. Unfortunately, the markers that have been developed to date for the enrichment of CBE- or ABE-edited cells exclusively employ nickase SpCas9[12–16], leaving the question open, whether they are sensitive enough to report on the activity of dead base editors. One of the objectives of our study was to develop a marker that enables high efficiency base editing by enriching dABE- and dCBE-edited cells, without deliberately nicking the DNA.

Another objective of the study focused on Cas9-dependent off-target edits of the base editors. Several base editors have been developed to eliminate some of the limitations of conventional base editing techniques. These improved variants involve mutant and deletion variants of deaminases, modified length of the linker between the nuclease and the deaminase, different orthologs and variants of Cas9 or Cas12a, additional copies of fused uracil glycosylase inhibitors (UGI), as well as changed architecture of base editors[9,11,17–26]. In fact, these variants have successfully increased the activity of base editors and altered their editing windows or their specificities towards the edited bases, as well as they have partially decreased associated Cas9-dependent or Cas9-independent off-target effects. A detailed description of these developed variants and their features can be found in reference[27]. However, while approaches to monitor genome-wide Cas9-dependent off-target modifications have been developed[28–30], methodologies to diminish them are less well established. One of the promising approaches is to apply increased fidelity variants of SpCas9[31–35]. These variants have been introduced in order to decrease the off-target editing of the nuclease version of SpCas9. These variants exhibit higher specificity and decreased activity in a target-dependent manner, seemingly trading efficiency for specificity[35–38]. Scientific literature in this area is lacking a

thorough assessment of the applicability of increased fidelity SpCas9 variants to decrease the mismatch tolerance of ABE and CBE. Thus, it has been designated as the second objective of our study.

Although a great number of base editor variants have been developed, it is rather difficult to get an overview of their features and the benefits these variants can offer relative to one another. The absence of simple and effective means to compare the performance of these base editors, in terms of on-target efficiency, tolerance for mismatches and relative activity at different positions of the extended editing-window on various sequences and in any cell of choice, hampers the exploitation of base editor variants to their full potential. To monitor the activity of base editors, usually, Sanger or next generation sequencing is applied[7,8,20,39]. Recently, a few approaches have been reported that allow the employment of fluorescence-based assays. These assays are based on the installation or alteration of a start or stop codon[15,16], or they rescue a disruptive amino acid and concomitantly recover a fluorescent signal[40]. Alternatively, a non-synonymous mutation in the chromophore of a fluorescent protein that induces fluorescence spectral change has also been explored as an option to monitor base editing activity[12,14]. Although these assays exploit clever strategies, they are limited to[12–14,40] and/or demonstrated[15,16] on few target sequences, exhibit high background signal[13] and/or rely on an integrated copy of the marker[15]. Their features are summarised in Supplementary Table 1. In this work, we report the development of Base Editor Activity Reporter (BEAR) and employ it to better understand whether and how increased fidelity SpCas9 variants can decrease the off-target effects of ABEs and CBEs.

## Results

**Development of the BEAR assay**. We aimed to develop an easy-to-perform and quick gain-of-signal fluorescent assay to monitor base editing activity with a plasmid-based format, that allows the use of numerous sequences and can be easily adapted to various cell types. The assay should report exclusively on the efficiency of base editing without being sensitive to potential indels generated by base editors. BEAR, the assay we designed in accordance with these requirements, is based on a split GFP protein separated by the last intron of the mouse *Vim* gene. The sequence of the functional 5' splice site (hereafter referred to as splice donor site) is altered to abolish splicing and thus GFP fluorescence, but both splicing and GFP fluorescence can be restored by applying base editors (Fig. 1).

This rationale could not be used by attempting to inactivate the canonical 'GT' splice donor site just by altering one base neither in the first position from 'G' to 'A' to be compatible with ABEs, nor in the second position from 'T' to 'C' for CBEs, as both 'AT' and 'GC' splice sites are known as very rare, but functional non-canonical splice sites in the human genome[41]. We also verified this by transfecting plasmids with these canonical ('GT') and non-canonical ('AT' or 'GC') splice donor sites into both N2a and HEK293T cells and then measuring the number of GFP positive cells afterwards (Supplementary Fig. 1).

To develop the assay, first, we wanted to interrogate the mutation tolerance of the splice donor site and its sequence environment. Next, based on this knowledge, we wanted to find out which inactive splice donor sites can be converted to an active splice site sequence the most efficiently by the action of base editors. In order to find appropriate inactive and active sequence pairs which fully diminish or support splicing, respectively, we have systematically modified the non-targeted nucleotide of the 'GT' splice donor site to 'AN' and 'GN' for ABEs (Fig. 2a) and to 'NC' and 'NT' for CBEs (Fig. 2b) corresponding to the inactive

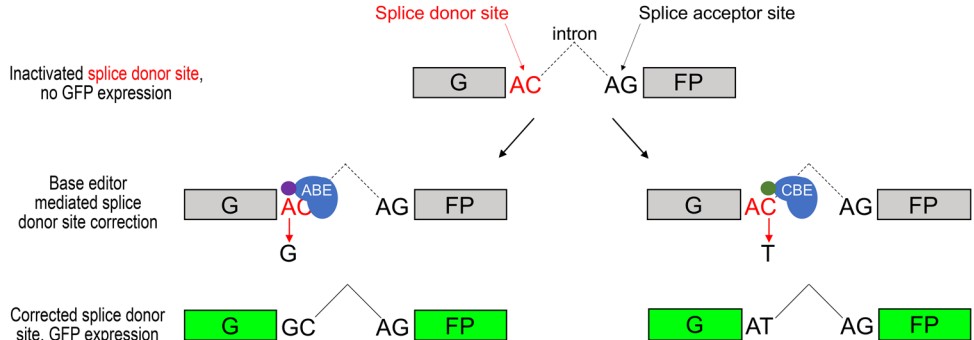

**Fig. 1 Principle of the base editor activity reporter (BEAR) assay.** BEAR consists of a split GFP coding sequence (green) separated by an intron of which splice donor site is altered to 'AC', resulting in a dysfunctional protein (grey). This inactive splice donor site can be rescued either by ABEs reverting the 'AC' splice donor site to 'AT' or by CBEs reverting the 'AC' splice donor site to 'GC', respectively. 'AT' and 'GC' are known to be functional non-canonical splice donor sites in the human genome.

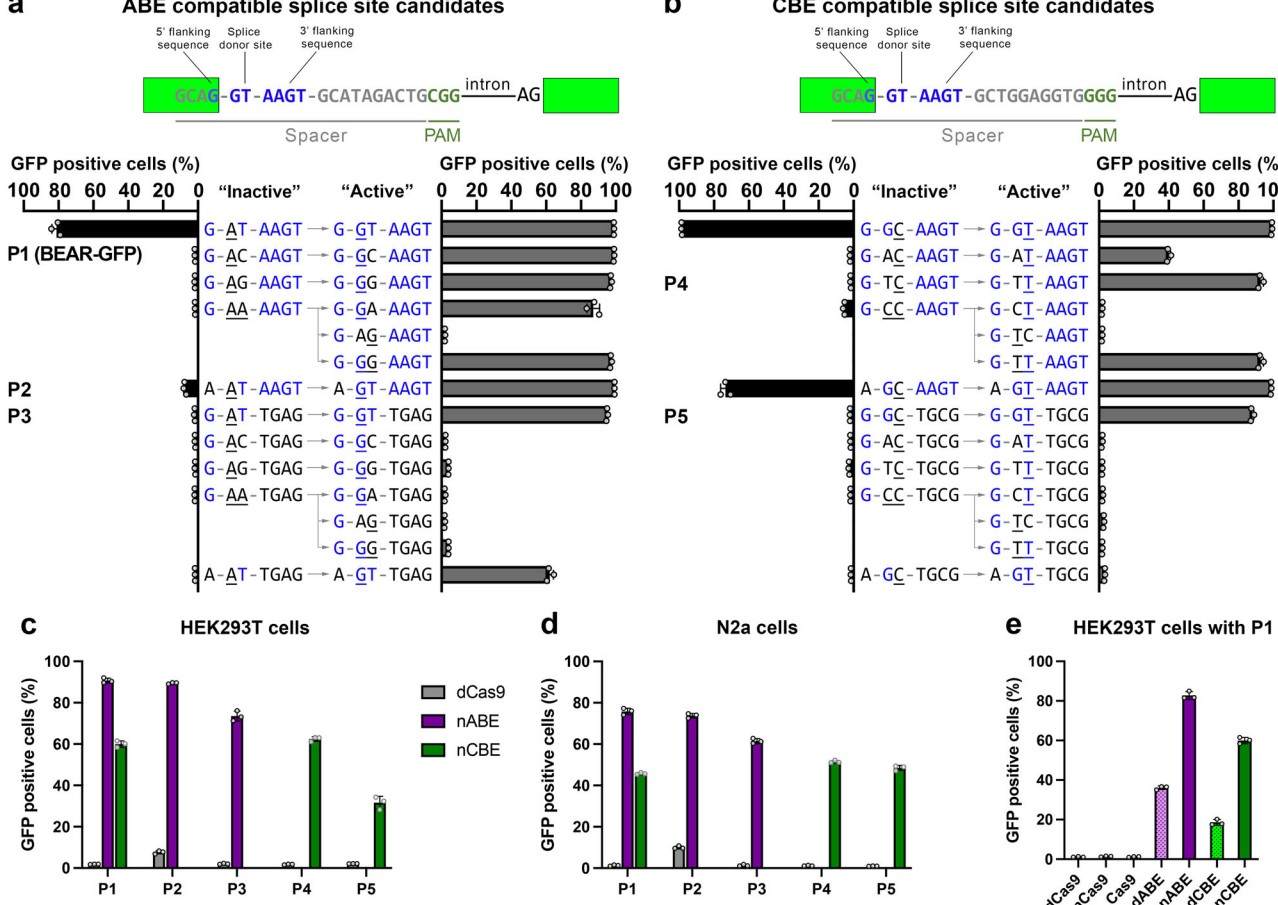

**Fig. 2 Splice site variants for identifying candidate BEAR sequences.** Flow cytometry measurements of GFP positive HEK293T cells, transfected with plasmids harbouring systematically altered splice sites (expected "inactive" sequences), which hold the possibility of being converted by ABE (**a**) or CBE (**b**) to sequences expected to be functional, "active" splice sites. BEAR plasmids with inactive and active splice sites were generated by molecular cloning, the latter representing the maximum fluorescence that can be achieved by base editing. The sequences between the column charts represent the intended "inactive" or "active" splice donor site and flanking sequence pairs. Letters highlighted in blue indicate the bases that correspond to the canonical 5' - G GT AAGT - 3' sequence (upper panels); the altered bases in the GT splice donor sites are underlined. Five sequence pairs (P1-P5) with minimal fluorescence for the inactive and maximal fluorescence for the active splice donor site were selected for further analyses as detailed in Fig. 2c-e. Note that the target sequence in the a and b panels are different. Flow cytometry measurements of GFP positive HEK293T (**c** and **e**) and N2a (**d**) cells co-transfected with a selected reporter plasmid harbouring an inactive splice site and base editor or control nuclease constructs as indicated in the figure. Columns represent means, +/- SD of three parallel transfections (grey circles).

and active splice sites, respectively. We also examined changing the 5' or 3' flanking sequences of the donor site in both the inactive and the active plasmids (here and throughout the manuscript active plasmids are the positive controls generated by molecular cloning to represent the maximum fluorescence that can be achieved by editing). The flanking sequences of the donor site are known to modulate the efficiency of the splicing process[42]. This exon-intron junction contains the 5' G and 3' RAGT flanking sequences (Fig. 2a, b), which have been reported to best enhance splicing[42]. Active and inactive constructs were transfected into both HEK293T and N2a cells without base editors, and the cells were analysed by flow cytometry (Fig. 2a, b; Supplementary Fig. 2a, b; respectively). These experiments revealed that altering only one of the bases of the splice donor site to any of the three other bases while keeping the flanking region intact was found to preserve fluorescence, whereas altering both bases of the splice donor site abrogated the fluorescent signals. Changing just one base only ceased fluorescence completely, when either the 3' or the 5' flanking sequence was altered as well. When both flanking regions were altered, even the canonical 'GT' splice donor site sequence was inadequate for efficient splicing and recovery of GFP fluorescence. These experiments revealed a few candidate combinations for which no detectable fluorescent signal is apparent with the inactive splice site sequence, but it is present in the case of the corresponding active sequences (Fig. 2a, b, Supplementary Fig. 2a, b).

Next, we tested whether base editors can indeed recover fluorescence by exploring some of the best candidate constructs identified in Fig. 2a, b and in Supplementary Fig. 2. Throughout the study, the codon optimised ABERA (shortened as ABE) and FNLS-CBE (shortened as CBE) variants of the adenine and cytosine base editors are used, respectively, described by Zafra et al.[22], unless indicated otherwise. The five selected plasmids (P1-P5 in Fig. 2a, b and Supplementary Fig. 2) were co-transfected with ABE and CBE, separately into both HEK293T and N2a cells, and the number of GFP positive cells were then measured. In the case of all selected constructs ABE and CBE could successfully recover fluorescence from 31% to 91% in HEK293T (Fig. 2c) and from 45% to 75% in N2a cells (Fig. 2d). The inactive splice donor site in P1 can be efficiently corrected by both ABE and CBE converting 'AC' to either 'GC' or 'AT', respectively, hence restoring GFP fluorescence. Since both ABE and CBE reach the same level with P1 as with the other best constructs (Fig. 2c, d), we could further examine both base editors on this one mutual inactive plasmid, hereafter called BEAR-GFP (Fig. 1).

Figure 2e shows that fluorescence is not recovered when the BEAR-GFP construct is targeted by a single nickase or a nuclease SpCas9, supporting that the method reports about base editing exclusively. We also found that the nuclease inactive base editor variants dABE and dCBE are also capable of correcting the splice donor site, however, with lower efficiency, as indicated by the recovered fluorescence signals of 36% and 18% for dABE and dCBE, respectively (Fig. 2e).

As an advantage, our method is not restricted to a few target sequences only. The intronic sequence downstream to the 3' flanking sequence can be varied without restrictions. This also allows the user to move the PAM sequence, and thus, the editing window, with respect to the base position to be edited (Supplementary Fig. 3a). Furthermore, the exonic part of the target sequence can also be altered by using BEAR with different fluorescent proteins (Supplementary Fig. 3b) or by shifting the position of the intron within the coding sequence of the protein (Supplementary Fig. 3c and d). Thus, even when the seven nucleotide-long flanking sequence part of the target sequence is

kept unaltered, more than a million of possible different target sequences can be examined using BEAR. Since even either the non-edited nucleotides of splice donor site or one of the flanking sequences can also be varied (Fig. 2a, b), our method allows the targeted base to be examined in almost any sequence context.

To see whether the efficiency of base editing of target sequences in a plasmid or in a genomic context is governed by the same factors, we have generated stable HEK293T cell lines harbouring either a split GFP or a split mScarlet protein, containing the exact same exons, intron and target sequence as the BEAR plasmids. When these cell lines were targeted by ABE and the corresponding sgRNA, fluorescence was efficiently recovered (Fig. 3a). Next, we have compared the BEAR-GFP plasmid with the BEAR-GFP cell line, in regard to their effects on the extent of fluorescence recovery, using 1 matching and 31 mismatching sgRNAs containing one or two consecutive mismatches at different positions (Fig. 3b). The assays on the cell line and on the plasmid yielded highly similar outcomes ($r = 0.89$, Supplementary Fig. 3e), indicating that the plasmid-based assay truly reflects the activities of ABEs on sequences in a genomic context.

To confirm that fluorescence recovery is the result of successful base editing, we have employed a fully matching sgRNA, and sgRNAs with one, two or three mismatches with ABE (Fig. 3c) or CBE (Fig. 3d) on the BEAR-GFP cell line, and monitored base editing activity by measuring the number of GFP positive cells, as well as by using Sanger sequencing to quantify editing with the EditR software[43]. The measured fluorescence intensity was found to be proportional to the level of actual base editing ($r = 0.98$ in case of ABE). In the case of CBE, all sgRNAs except the one with a three-base mismatch gave maximum signal in both contexts. This different mismatch tolerance of ABE and CBE is examined in more detailed later in this study. Sequencing has also revealed that in the case of ABE not only the splice donor site sequence, but also a bystander adenine has been edited to a certain extent (Fig. 3e). Constructing and testing the corresponding inactive and active plasmids have proved that editing the second, bystander 'A' with or without the adenine of the splice donor site sequence does not decrease or increase GFP fluorescence, respectively (Fig. 3f). In the case of CBE, no bystander nucleotides were edited, but the targeted cytosine was converted also to guanine, although to a smaller extent (Fig. 3g), accordingly to previous reports with several other target sequences[21,39,44]. By constructing the corresponding active plasmids, we have verified that the increase in fluorescence is derived from the intended editing of 'AC' to 'AT' only, without a contribution from 'AC' to 'AG' editing of the splice donor site (Fig. 3h). Altogether, these data support that the BEAR method gives a reliable account of the activities of a base editor.

**Increased base-editing yield without nicking the target DNA.** Since BEAR is able to detect the activity of nuclease inactive base editors (Fig. 2e), we tested whether it could be used as a marker for those cells, in which efficient base editing had occurred, in order to increase the yield of base editing without deliberately nicking the DNA. First, as a proof of principle, we used the BEAR plasmids in combination with genome-integrated copies of different BEAR colours to see if they indeed can label cells, in which base editing events had taken place at genome-integrated targets. We co-transfected the BEAR-GFP plasmid with dABE and the corresponding sgRNAs into the BEAR-mScarlet cell line, and we have found that dABE restored mScarlet fluorescence in 20% of the cells (Fig. 4a). Thirty-one percent of the cells in the trans-fected population, (which is gated to a blue-colour transfection marker), exhibited mScarlet fluorescence, and 51% of the cells showed fluorescence for both mScarlet and GFP, indicating that the cells that are active in processing the A-to-G base conversion

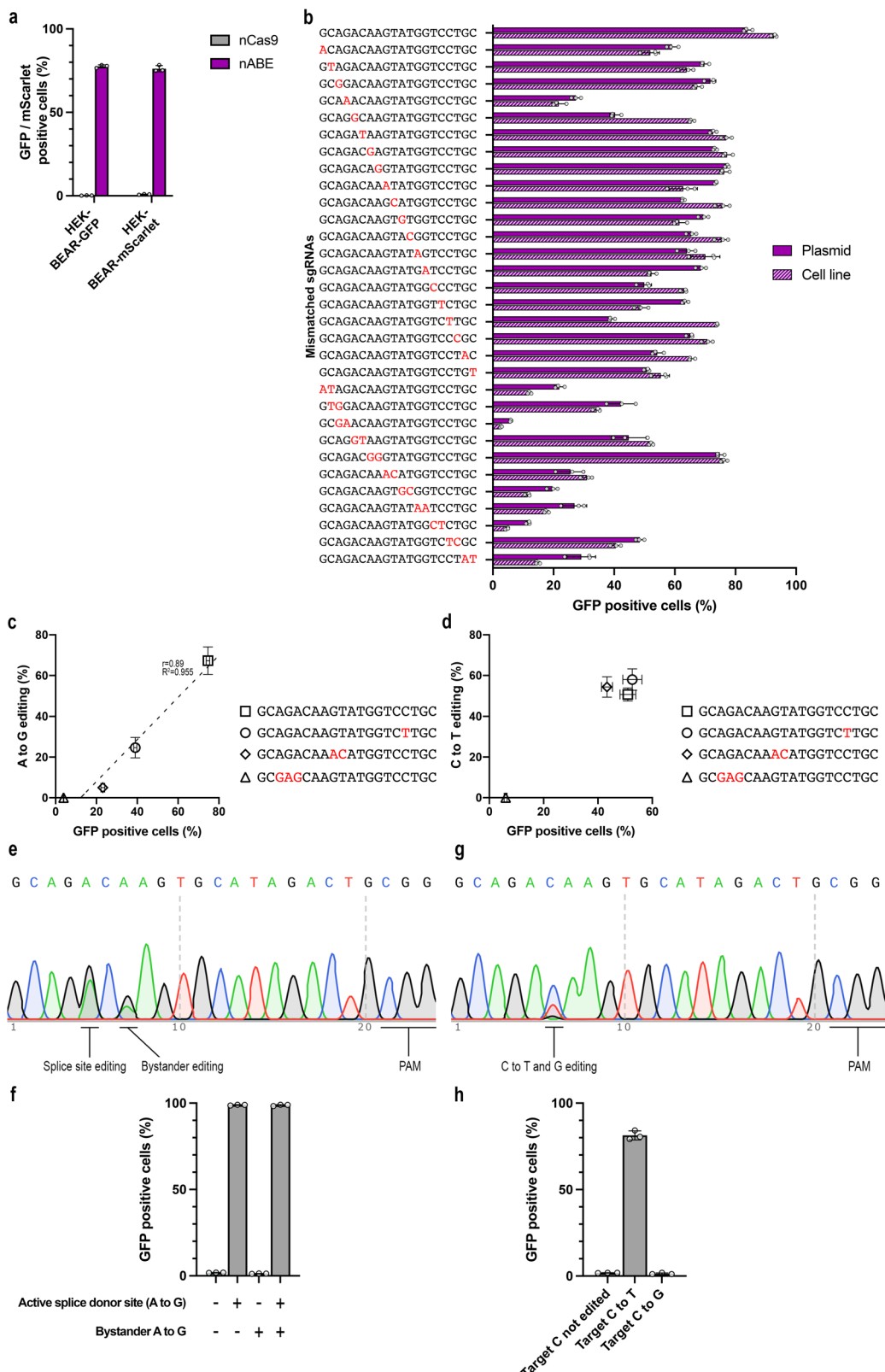

on the plasmid are also efficient on the genomic DNA (Fig. 4a). We also co-transfected the BEAR-mScarlet plasmid with dABE into the BEAR-GFP cell line. In this experiment BEAR-enrichment (i.e., gating to the mScarlet positive population,) has increased the percentage of edited cells from 22% to 45%, highly exceeding the enrichment that we measured in the transfected population (30%; Fig. 4a).

To make enrichment experiments simpler, we constructed a BEAR-GFP plasmid which also bears its targeting sgRNA on the same plasmid. We named this construct BEAR-GFP-2in1 and used it in subsequent experiments to enrich base editing on endogenous genomic targets (HEK site 1-4, CCR5, FANCF site 2 and SCN5a, Fig. 4b). Employing dCBE (Fig. 4c–e) and dABE (Fig. 4g–i), we base edited 5 genomic targets using each, while

**Fig. 3 Validation of BEAR in genomic contexts.** Flow cytometry measurements of fluorescence (**a–d**, **f**, **h**) and Sanger sequencing (**c–e**, **g**) revealed that the GFP/mScarlet fluorescence recovery in the BEAR sequence is indeed the result of the intended base editing. **a** Experiment in BEAR-GFP and BEAR-mScarlet cell lines in which the genome-integrated copies of the BEAR plasmid are targeted by ABE and by nCas9 as a negative control. Columns represent means ± SD of three parallel transfections (grey circles). **b** The integrated copy (BEAR-GFP cell line, striped purple) and transfected BEAR plasmid (purple) were targeted by ABE using the same sgRNAs, one matching and 31 mismatching in one or two positions as indicated by red letters. Comparison of GFP signal recovery showed a correlation of Pearson's $r = 0.89$ (Supplementary Fig. 3e). Columns represent means ± SD of three parallel transfections (grey circles). **c**, **d** Scatter plots corresponding to the GFP positive cells and base editing quantified by EditR resulted from the BEAR-GFP cell line interrogated by ABE (**c**) or CBE (**d**) using sgRNAs, one matching and three mismatching in one, two or three positions as indicated with red letters in the target sequences. Each point on the scatter plots represent means, error bars represent ± SD of three parallel transfections. **e**, **g** Chromatograms of Sanger sequencing of the target region of the BEAR-GFP cell line. **e** Sequencing of the ABE-edited BEAR-GFP cell line revealed that a bystander adenine was edited in the 3′ flanking sequence. **f** Testing the corresponding active plasmids proved that editing the bystander adenine alone or together with the targeted adenine does not influence GFP fluorescence. Columns represent means ± SD of three parallel transfections (grey circles). **g** In the case of the CBE-edited cell line, no bystander edits were detected. However, a portion of the edited C was converted to G instead of T. **h** Testing the corresponding active plasmids proved that changing the edited C to G does not influence GFP fluorescence. Columns represent means ± SD of three parallel transfections (grey circles).

also co-editing the BEAR-GFP-2in1 plasmid in the same cells as a marker for efficient base editing. GFP positive cells were cell sorted and base editing was quantified in the population by next-generation sequencing (NGS) and compared with not enriched cells or with cells sorted for the transfection marker. For comparison, nABE and nCBE (containing nickase SpCas9) editing was also assessed (without enrichment) on the same target sites.

In case of dCBE we achieved a 2.8–4.8-fold enrichment on the five targeted sites which corresponds to approximately the same level of base editing as nCBE without enrichment (Fig. 4c). For comparison, a 1.1–1.3-fold enrichment was achieved when using the transfection marker. We also monitored indel formation in the base edited samples (Fig. 4d) and as expected, BEAR enrichment of dCBE editing resulted in a 3.1- to 30.1-fold increase in specificity (compared to nCBE), defined here as on-target editing % / indel % (Fig. 4e). Using dABE, we achieved 1.3–2.9-fold enrichment on the five targeted sites which corresponds to approximately the same level of base editing as nABE without enrichment (Fig. 4f). For comparison, a 1.2-fold enrichment was achieved on all targeted sites when using the transfection marker. In agreement with the literature[45], nABE yielded an order of magnitude less indels compared to nCBE (Fig. 4g). With enriching dABE edited cells this low amount of indel formation could even be lowered to the detection limit of next generation sequencing (0.05%). Compared to nABE, BEAR enrichment of dABE editing resulted in a 1.1- to 21.9-fold increase in specificity (Fig. 4h). We also enriched nABE editing on the same target sites and on average, we found a 1.6-fold higher percentage of base edited sequences (Supplementary Fig. 4a) while maintaining low (below 0.8%) indel formation (Supplementary Fig. 4b). Altogether, these experiments indicate that BEAR-enrichment yields base edited genomic targets with about nickase-level (nABE and nCBE) efficiency while preserving the low indel background of dABE and dCBE.

**On-target activity of increased fidelity base editors.** Several studies reported on CBE showing higher or similar mismatch tolerance compared to ABE, which results in various Cas9-dependent off-target effects[10,30,46,47]. Applying increased fidelity variants may seem to be a plausible approach to decrease the Cas9-dependent off-target effects of base editors. However, no study provides a thorough assessment of this alternative, although a few attempts of combining an increased fidelity variant with a base editor are reported in the literature[20,30,48–51]. To get a more comprehensive understanding of these effects, using BEAR, we compared the activity and mismatch tolerance of CBE and ABE containing six increased fidelity SpCas9 variants: eSpCas9, SpCas9-HF1, HypaSpCas9, HypaR-SpCas9 (i.e., HypaSpCas9

which also contains the R661A mutation), evoSpCas9 and HeFSpCas9[31–35] (Fig. 5). Using these increased fidelity SpCas9 variants six increased fidelity ABEs (e-ABE, HF-ABE, Hypa-ABE, HypaR-ABE, evo-ABE and HeF-ABE) and six increased fidelity CBE variants (e-CBE, HF-CBE, Hypa-CBE, HypaR-CBE, evo-CBE and HeF-CBE) were constructed, respectively.

Considering that the 'AC' splice donor site sequence in the BEAR-GFP plasmid can be edited by both ABEs and CBEs (Fig. 1e), they can be compared on the same targets, by using the same sgRNAs. Accordingly, we compared their on-target base editing activities on 34 targets in which the splice donor site and flanking regions, as well as their distance from the PAM sequence, was kept fixed and only the PAM proximal 10 nucleotides were varied. Thus, in case of both base editors, the sequences in their editing windows and the bases surrounding the edited bases were the same in all targets. Neighbouring (+/-1) nucleotides can strongly influence the efficiency of base editing; 'GAC' and 'ACA' employed here for ABE and CBE, respectively, have been shown to be associated with medium level activities for both base editors[52]. Lacking data suggesting otherwise, we expected that the differences between the 34 target sequences (in the PAM proximal 10 nucleotides) should primarily affect the interactions between the fused SpCas9 nuclease partner of the base editors and the targets, thus, this experimental design specifically allowed the study of how the binding and cleavage propensities of SpCas9 variants affect the activities of base editors.

The activity of base editors containing both the nickase (nCBE and nABE) and the inactive SpCas9 (dCBE and dABE) was measured so that we could then compare it with the activity of the variants containing increased fidelity SpCas9 variants. The results illustrated in Fig. 5a indicate that nABE is highly active on all 34 targets with 73% mean activity (its efficiency ranging from 62% to 89%). dABE was found to be less active with a 24% mean activity. In theory, the activity profile of dABE is influenced by the sequence specificity of both the TadA deaminase and the binding of SpCas9. In contrast, the activity profile of nABE is additionally influenced by the nicking activity of SpCas9, which aims to bias the repair system into correcting the mismatching bases of the unedited strand, thus to increase editing efficiency[7]. The activity profile of dABE and nABE shown in Fig. 5a indicates a weak correlation ($r = 0.29$; Supplementary Fig. 5a), suggesting that the nicking activity of SpCas9 in nABE substantially alters the relative efficiency of nABE compared to dABE on these target sequences.

In respect of increased fidelity SpCas9 variants former studies have shown that these nucleases have a trade-off between efficiency and fidelity, and can be ranked according to their average activities, with evo- and HeFSpCas9 showing much lower average activities than the rest of the increased fidelity variants[35–38]. In our experiments, increased fidelity variants of

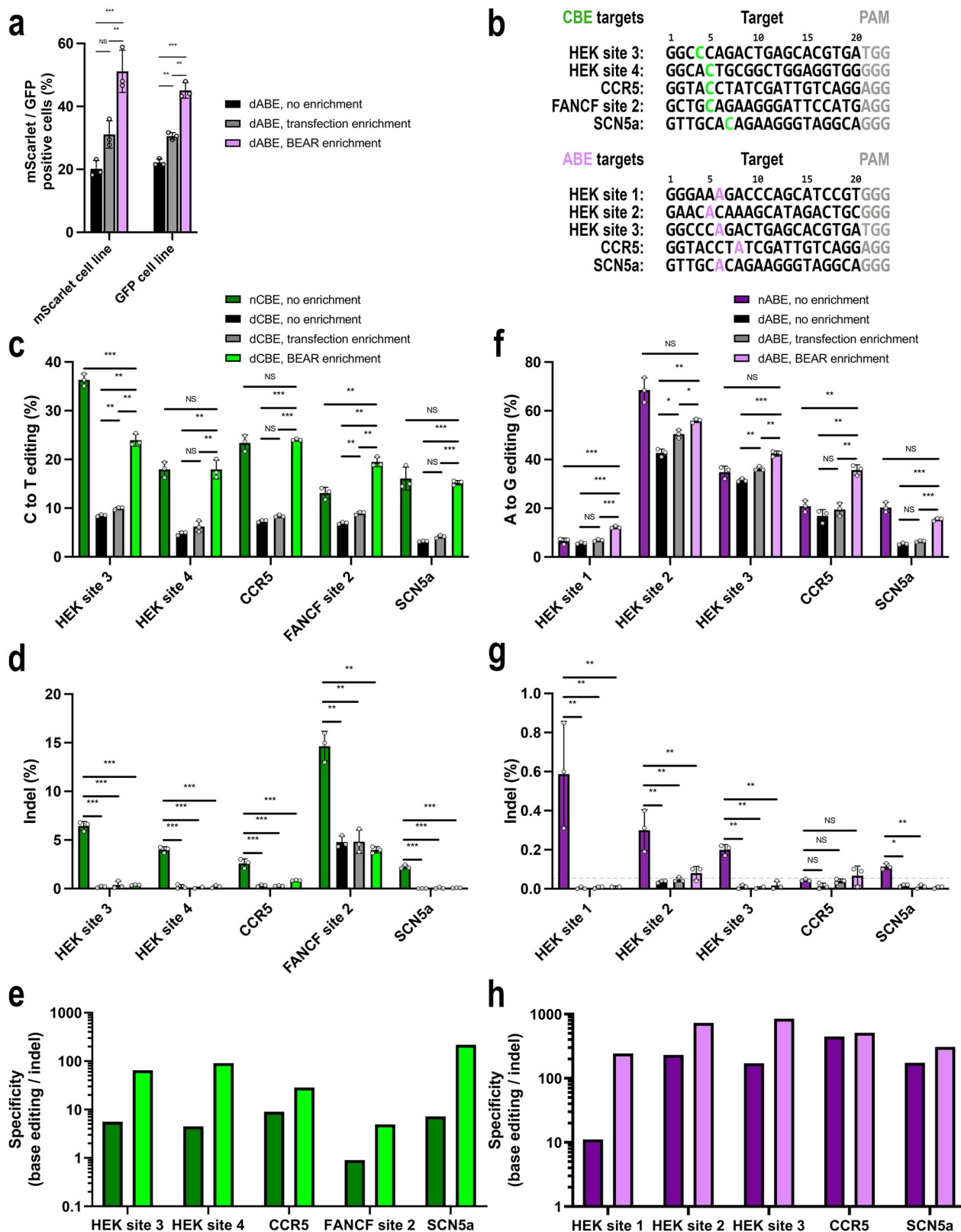

ABE exhibited decreasing activity starting from ABE towards HeF-ABE, the latter showing minimal activity (4% in average; Fig. 5a). This result resembles the activity decrease seen in the case of increased fidelity nucleases[35–38].

The same experiments with CBEs showed that nCBE is less active on these 34 targets (its average editing activity is 50%) and exhibits higher sensitivity for sequence variations: its efficiency ranged between 26% and 69% (Fig. 5b). dCBE is considerably less active (with an average activity of 12%) and its efficiency varied from 5% to 21% (Fig. 5b). Their activity profiles of nCBE and dCBE correlate ($r = 0.51$, Supplementary Fig. 5b) more than the activity profiles of the ABEs, raising the interesting possibility that the nicking activity has a weaker relative influence on the sequence dependence of CBE than on that of ABEs.

**Fig. 4 Enrichment of dABE and dCBE base edited cells with BEAR, matches the efficiency of nABE and nCBE while creating less indels. a** Either the BEAR-mScarlet or the BEAR-GFP plasmid and dABE were co-transfected into the BEAR-GFP or BEAR-mScarlet cell line and the cells were analysed by flow cytometry gated for GFP or mScarlet positive cells, respectively. Cells without additional gating: no enrichment (black bars) with additional BFP gate: transfection enrichment (grey bars), with additional BEAR-plasmid colour gate: BEAR enrichment (purple bars). To monitor transfection efficiency a blue fluorescent protein (BFP) was placed onto the sgRNA-expressing plasmid. **b** The target sequences of the endogenous genomic targets edited with CBE or ABE variants in panels **c** and **f**. Bases edited most efficiently are coloured and their efficiencies are depicted on panels **c** and **f**. For all other edited bases and editing efficiencies see the Source Data file. The BEAR-GFP-2in1 plasmid and endogenous genomic targets were co-edited by dCBE (**c–e**) and dABE (**f–h**) and analysed by NGS. Edited cells were sorted to 3 fractions: all cells (no enrichment, black), BFP positive cells (transfection enrichment, grey), and cells with GFP positivity representing base editing enriched cells (dCBE – light green, dABE – light purple). In all experiments nCBE (dark green) and nABE (dark purple) edited cells were monitored without enrichment as controls. Base editing and indel formation was quantified from the same samples in the case of dCBE (**c, d**) and dABE (**f, g**) edited cells. The specificity (base editing % / indel %) is displayed for nCBE and dCBE (**e**) as well as for nABE and dABE (**h**). Columns represent means ± SD of three parallel transfections (grey circles). When indel % was lower than the detection limit of NGS (dashed grey line, 0.05%), specificity was calculated with 0.05% indel to avoid falsely high specificity values. Differences between samples were tested using one-way ANOVA. NS:$p>0.05$, *$p<0.05$, **$p<0.01$, ***$p<0.001$. For source data and exact $p$-values see the Source Data file.

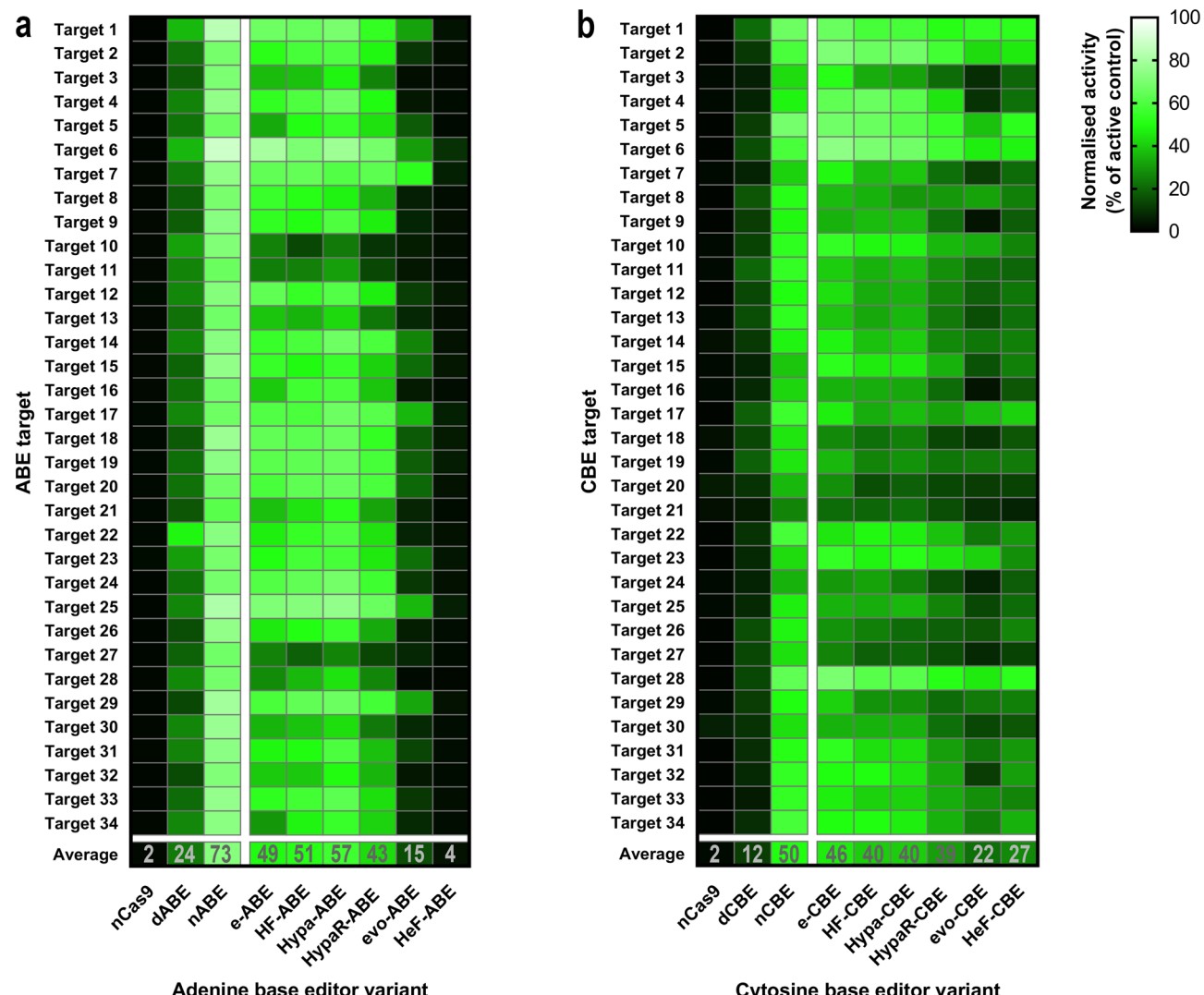

**Fig. 5 On-target activities of various increased fidelity ABE and CBE variants on 34 target sites. a, b** The heatmap shows the mean on-target activity deriving from three parallel transfections normalised to the active splice donor site plasmid. A total of 34 target sequences, differing in their PAM proximal 10 nucleotides, were used to test dABE, nABE and 6 increased fidelity ABEs (**a**) as well as dCBE, nCBE and 6 increased fidelity CBEs (**b**) as indicated in the figure. As a negative control, nCas9 was used with all 34 targets. For source data see the Source Data file.

A decreasing effect of increased fidelity mutations, from e- to evo- and HeF-CBE variants, on the average activities of CBE is also apparent, although this decrease is much less prominent than it is in case of the ABE variants: their average activity decreases from 50% to 22% compared to the 73% to 4% decrease seen with ABEs (Fig. 5b).

These experiments suggest that the impact of increased-fidelity mutations is more evident in ABE than in CBE variants.

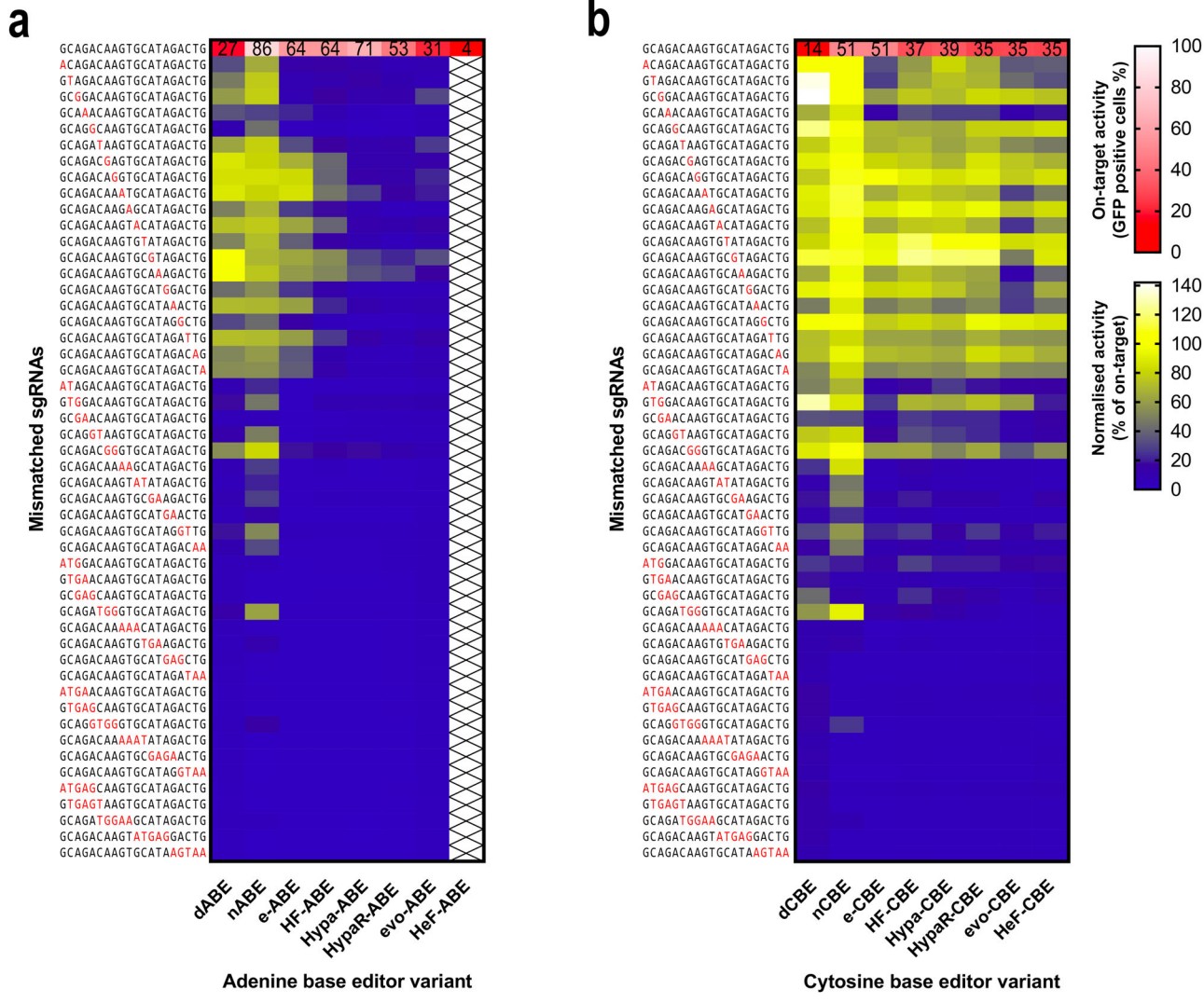

**Fig. 6 Off-target activities of various ABE and CBE variants with 50 mismatched sgRNAs.** Mismatch tolerance of ABE (**a**) and CBE (**b**) and their increased fidelity variants were compared utilising the same matching sgRNA (target 1 in Fig. 5) and 50 sgRNAs mismatching in one, two, three, four or five positions as indicated as red letters. Blue and yellow heatmaps show the mean normalised activity (off-target/on-target) derived from three parallel transfections. White and red heatmaps show the on-target activity (mean rates of GFP positive cells) derived from three parallel transfections. For source data see the Source Data file.

**Mismatch tolerance of increased fidelity ABE and CBE variants.** We compared the mismatch tolerance of ABE (Fig. 6a) and CBE (Fig. 6b) along with their increased fidelity variants employing 50 mismatching sgRNAs (using target 1 from Fig. 5), in which the position of one to five consecutive mismatches were varied systematically along the full length of each sgRNA. Investigating nABE, we found that it tolerates sgRNAs containing one or two mismatches in all the positions examined, with an average of 71% and 37% normalised activity (% of on-target activity), respectively (Supplementary Fig. 6a). dABE exhibited slightly higher fidelity than nABE, which is more apparent with the sgRNAs containing two mismatching positions (normalised average activity of the 2 mismatching sgRNAs of 15%, Supplementary Fig. 6a). The mismatching profiles of nABE and dABE show a strong correlation ($r = 0.88$, Supplementary Fig. 6b), which is interesting, since the off-target effects of the active and inactive forms of SpCas9 have been reported to differ substantially[53]. Considering this, we expected a weaker correlation, similar to the weak correlation between the on-target activities of nABE and dABE.

Regarding the increased fidelity ABE variants, five of them were tested on the BEAR-GFP plasmid (target 1 from Fig. 5), employing the same 50 mismatching sgRNAs. HeF-ABE was excluded from these experiments due to its low on-target activity. Increased fidelity mutations were found to decrease the mismatch tolerance of ABE (Fig. 6a). The fidelity of the same SpCas9 nuclease variants have been reported to increase to a great extent from eSpCas9 to evo- and HeFSpCas9[35–37]. Remarkably, these fidelity increases are also evident in the mismatch tolerance of the ABE variants when sgRNAs mismatching in one position are employed (Fig. 6a, Supplementary Fig. 6a). In contrast, with almost all sgRNAs containing two or more mismatches, each increased fidelity ABE variant was found to exhibit only background-level activities. Interestingly, increased fidelity ABE variants exhibit higher specificities on this target than dABE (Fig. 6a).

The mismatch tolerance of the CBE variants was also tested using the same 50 mismatching sgRNAs (Fig. 6b). nCBE tolerates one or two mismatches in all the positions examined, with an average normalised activity of 100% and 62%, when the sgRNAs

contain mismatches in one or two positions, respectively (Supplementary Fig. 6c). In turn, dCBE exhibits slightly higher fidelity, which is more apparent with the sgRNAs containing two mismatching positions (44% normalised average activity). The mismatch profiles of nCBE and dCBE show a strong correlation ($r = 0.87$, Supplementary Fig. 6d), which is similar to that seen with nABE and dABE (Supplementary Fig. 6b).

Regarding the increased fidelity CBE variants, all six of them reached sufficiently high on-target activity on the BEAR-GFP plasmid, thus all six have been investigated with the previous set of 50 mismatching sgRNAs. Although an overall increase in specificity towards the highest fidelity evo- and HeF-CBE was evident (Fig. 6b), this effect is much less prominent than it is in the case of the increased fidelity ABE variants (Fig. 6a). Compared to ABEs, increased fidelity CBE variants exhibited lower specificity. Specifically, while ABE variants showed, 4–6% and 2–5% of normalised average activity, with sgRNAs mismatching in two or three positions, respectively (Supplementary Fig. 6a), CBE variants exhibited 16–27% and 6–12% of normalised average activity with the respective mismatching sgRNAs (Supplementary Fig. 6c)

To see whether these observations are specific to the target examined or are more general characteristics of these base editor variants, we investigated the mismatch tolerance of ABE and CBE variants on three additional targets (targets 2, 6 and 17; Supplementary Fig. 7a–c) using the same approach. These experiments confirmed the conclusions drawn from our previous findings shown in Fig. 6. Namely, (i) CBE is more tolerant to mismatches than ABE is, although ABE also shows a considerable target-dependent mismatch tolerance. (ii) Their activity profiles investigated using the same set of mismatching sgRNAs show strong correlations ($r = 0.93$–$0.96$, Supplementary Fig. 7d), arguing that their mismatch tolerances are primarily influenced by the specificities of SpCas9 cleavage (or being more accurate, nicking) activity as seen also with target 1 in Fig. 6 earlier. (iii) The effect of variants with higher fidelity SpCas9s is more prominent in the case of the ABE variants than in the case of the CBE variants, indicating that increased fidelity mutations decrease the mismatch tolerance of ABE more effectively than that of CBE.

We proposed that the sharp contrast between the differences of the specificities of ABEs and CBEs seen here may be related to the difference in the activity of their deaminases. Recently, an adenine base editor variant called ABE8e was created by Liu and co-workers that showed higher single strand DNA deaminase activity[54,55]. To test our hypothesis, we examined the on-target activity and mismatch tolerance of nuclease inactive dABE8e, ABE8e (hereafter, we call it as nABE8e to emphasize the presence of the nCas9 partner) and its increased fidelity variants. The results, illustrated in Fig. 7a, indicate that nABE8e is highly active on all 34 targets with 76% mean activity (its efficiency ranges between 62% and 84%). dABE8e was found to be less active with 31% mean activity. The activity profile of dABE8e and nABE8e, shown in Fig. 7a, indicates a strong correlation ($r = 0.72$; Supplementary Fig. 8a), that is much stronger than seen before for ABEs (Fig. 5a, Supplementary Fig. 5a) and is closer to but still stronger than for CBEs, suggesting that the nicking activity of the SpCas9 partners have smaller influence on its activity. The overall on-target data across the 34 targets and with the 6 variant partners shows higher similarity to CBE, than for ABE ($r = 0.67$ and $r = 0.36$, respectively, Supplementary Fig. 8b–d). The most apparent change is that the HeF- and evo-ABE8e variants exhibit higher average activities than dABE8e, which is also more similar to CBE, than that of ABE (Fig. 5).

The mismatch tolerance of ABE8e is much greater than that of ABE and even that of CBE (Fig. 7b). In the case of both dABE8e

and nABE8e, at least 3 mismatches in the sgRNA are necessary to decrease their activity to levels lower than with a perfectly matching sgRNA. Increased fidelity variant partners have an overall but modest decreasing effect on its mismatch tolerance, still exhibiting some activity with sgRNAs containing 4 or even 5 mismatches. ABE8e has extremely high activity and it was even reported to edit adenines that are not in optimal position in and around the target sequences[54,55]. To assess the effect of this promiscuous activity on the splicing and on the GFP function, we applied ABE8e to the BEAR-GFP cell line. Sanger sequencing revealed that the two adenines of the 3' AAGT sequence following the splice donor site were also converted to guanines; the first one in 66%, the second one in 16% of the amplicons. We generated the corresponding inactive and active plasmids containing the A to G edits (Supplementary Fig. 9b). These bystander edits did not have a considerable effect on the recovery of the functioning GFP protein. ABE8e also edited the last glutamine of the exon to arginine (CAG to CGG); interestingly, this mutation led to a complete loss of green fluorescence even in the context of the full protein constructs (i.e., without intron). The efficiency of this latter A-to-G bystander edit is substantial, about 66% of the on-target edit; however, it is still not too high to fully abolish the recovery of BEAR-fluorescence in the individual cells, where more copies of the BEAR sequence are edited.

Altogether, these results support our suggestion that the differences between the specificities of ABE and CBE are caused by the difference in the activities of their deaminases.

## Discussion

Base editors have been developed to edit the genome without deliberately causing DNA double-strand breaks. However, the application of nuclease inactive base editor variants is associated with low editing efficiency which limits their use immensely[7]. Thus, instead of these nuclease inactive base editor variants, base editors are generally applied with concomitant nicking of the targeted strand which results in considerably more efficient genome modifications, but on the other hand leads to increased incidence of indels[7,8,11]. As an alternative, we suggest employing a plasmid-based, fluorescent tool named BEAR (Base Editor Activity Reporter), which makes it possible to achieve the high editing yield attainable by nickase base editors without deliberately nicking the DNA, and thus, without generating considerable amounts of indels by the enrichment of cells edited with dABE or dCBE. This strategy is especially beneficial for CBE variants that execute edits with higher concomitant indel formation. As such, BEAR provides a unique solution for base editor applications, where indels generated by DNA nicking are not well tolerated (Fig. 4).

The experimental data provided here demonstrate the versatility of BEAR for comparing base editors with various features (Fig. 5 and Fig. 6) to select the best one for a particular task, as well as for facilitating the development of base editor variants with improved properties. The features of BEAR are compared to other reporter systems in Supplementary Table 1. One of its greatest advantages is its compatibility with millions of different target sequences. Its versatility in terms of accepting target sequences allows the comparison of the efficiency of base editing in different positions of the editing window, as well as the comparison of how neighbouring nucleotides of the edited bases affect the editing efficiency. These features of BEAR is demonstrated in Fig. 2 and Supplementary Figs. 2 and 3. However, in most of the experiments of this study, to assess how the nuclease activity of SpCas9 affects base editing, we kept the PAM distal region (the region surrounding the targeted nucleotide) constant, as these sequences are considered to have the most effect on deaminase function[52,56].

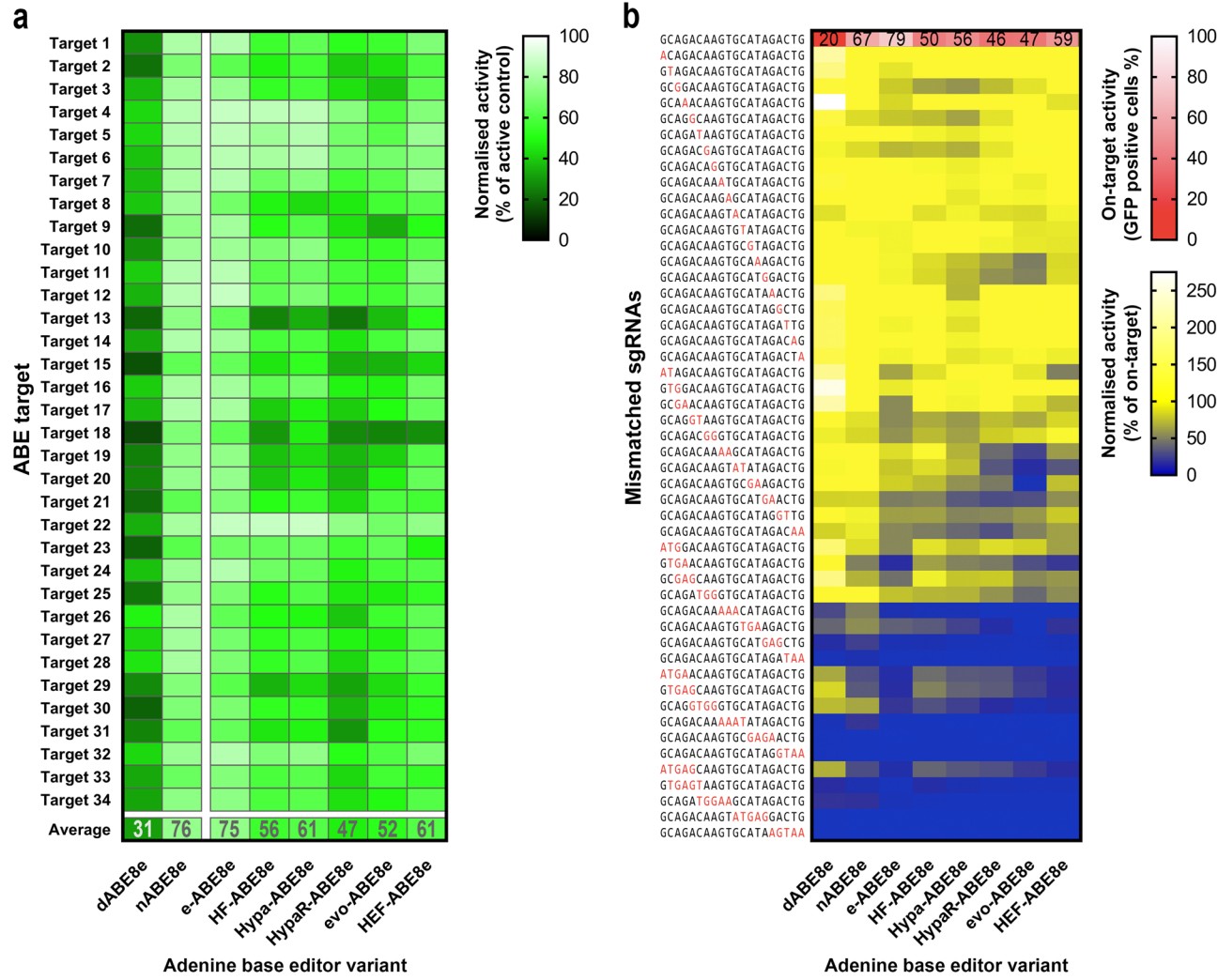

**Fig. 7 On- and off-target activities of various ABE8e variants. a** The heatmap shows mean on-target activity of three parallel transfections normalised to the active splice donor site plasmid. A set of 34 previously used targets (Fig. 5) were used to determine on-target editing efficiencies of dABE8e, nABE8e and 6 increased fidelity ABE8e base editors. **b** Mismatch tolerance of ABE8e and its increased fidelity variants were compared utilising the exact same matching sgRNA (target 1 in Fig. 5) and 50 sgRNAs mismatching in one, two, three, four or five positions as indicated as red letters. Blue and yellow heatmaps show the mean normalised activity (off-target/on-target) derived from three parallel transfections. White and red heatmaps show the mean on-target activity (percentages of GFP positive cells) derived from three parallel transfections. For source data see the Source Data file.

An interesting finding from our experiments is the low activity of the increased fidelity variant HeF-ABE, but not HeF-CBE (Fig. 5). Fidelity-increasing mutations have been shown to be associated with somewhat reduced nuclease activity: these nuclease variants are reported to pass through the quality checkpoints less efficiently[33,57] than the wild type SpCas9[58–60] during target cleavage, whereas their binding to the target DNA is largely unaffected[33,35]. This may be manifested in lower cleavage activity on off-targets, when there are mismatches between the spacer and the target DNA sequences, and sometimes on on-target sequences as well[31–35,38,50]. With such non-cleavable targets, the variants separate the DNA strands in the PAM-proximal region upon binding to the target, however, the mutations inhibit the effective full-length separation of the two strands of the target DNA, and thus they also prevent the formation of the cleavage-competent conformation of SpCas9[57,60,61]. HeFSpCas9, which has the highest fidelity and lowest average cleavage activity among these variants, binds to the targets of the wild type SpCas9 without being able to cleave most of them, unlike the wild type nuclease[58–60]. It fails to effectively separate the strands of the target DNA at full length and to acquire the cleavage-competent

conformation[33,35,57]. Based on these considerations, we expected that HeF-ABE, which lacks a nicking activity, would show a rather dABE-like activity on many targets and a dABE-like activity profile. Interestingly, our findings did not support this anticipation. The average activities of both evo- (the other highest fidelity variant) and HeF-ABE (15% and 4%, respectively, Fig. 5a) are less than that of dABE (24%), and their activity profiles hardly correlate with that of dABE (r = 0.16 and 0.29, respectively, Supplementary Fig. 10). In fact, dSpCas9 in dABE separates the strands of the target DNA at full length[61], facilitating the action of the deaminase, which is only active on single-strand DNA[8]. However, when increased fidelity nucleases are simply bound to the DNA without being able to cleave or nick the target, they separate the two strands of the target DNA at its seed region only, while the PAM distal region is separated less efficiently, rather transiently[57,60,61]. We speculate that in such cases, the deaminase is able to proceed in dABE, but not in HeF-ABE. Interestingly, the average activities of evo- and HeF-CBE do not fall below that of dCBE (22% and 27% versus 12%, respectively, Fig. 5b), suggesting that the effective, full-length separation of the two strands of target DNA is less critical for CBE than for ABE. Compared to

ABE, CBE may have a greater activity on targets where the separation of the PAM-distal region is rather transient in nature, likely due to the higher deaminase activity of APOBEC in CBE than that of the evolved TadA in ABE[27]. Liu and colleagues have developed a new version of ABE, ABE8e which exhibits a 1,000-fold higher deaminase activity and does not require the full-length separation of the two target DNA strands; it is active even on a transiently separated DNA strand, upon SpCas9 binding to a non-target PAM motif[54,55]. Our experiments with ABE8e involving the same 34 targets, 50 mismatching guides and 6 increased-fidelity nucleases (Fig. 7) show that HeF-ABE8e indeed has higher average activity than dABE8e, similarly to CBE (shown in Fig. 5). This supports our above interpretation, that the higher deaminase activity in CBE leads to the editing of targets at which increased-fidelity nuclease variants are not able to separate the two DNA strands at full-length, rather just transiently.

This observation also provides an explanation for the higher off-target propensity of CBE. Mismatches between the spacer and the target sequences have been described to reduce the cleavage activity of SpCas9 via a mechanism that also involves less efficient full-length separation of the target DNA strands[55,57,60,61]. Thus, ABEs are more sensitive to mismatches interfering with SpCas9 activity to an extent that inhibits the full-length separation of the strands of the target DNA. By contrast, CBE, which exhibits higher deaminase activity[27], can also act on some of the off-target sequences that are rather just transiently separated in the PAM distal region of the target DNA. These effects are even more apparent for ABE8e, which exhibits such high deaminase activity that it is able to deaminate adenines in transiently separated DNA strands upon binding to PAM sequences without a cognate protospacer sequence[54,55].

These interpretations are also in line with the following observations: (i) the pattern of gradually increasing fidelity and decreasing average activity of increased fidelity nucleases from enhanced SpCas9 to HeFSpCas9 nucleases are evident in the activities of increased fidelity ABE variants (Fig. 5a). In fact, these similarities are expected, if we consider that ABE variants preferentially work on those sequences on which the corresponding nuclease variants show full-length strand separation, and thus, show cleavage activities. These cleavage activity profiles of the increased fidelity SpCas9s are less discernible when we use CBE or ABE8e variants, which are less sensitive to the effective separation of the target DNA strands. (ii) Increased fidelity variants decrease the off-target activity of ABE more efficiently than that of CBE or ABE8e (Fig. 6; Fig. 7; Supplementary Fig. 7). (iii) The activity and specificity of ABE8e and its variants are more similar to CBEs than to ABEs.

This understanding of the actions of ABE and CBE variants suggests that the Cas9-dependent off-target propensity of ABE editing can be effectively diminished by the application of increased fidelity variants. It also suggests a strategy for CBE editing: fine tuning the deaminase activity of APOBEC to match that of the evolved TadA would make CBE combinations with increased fidelity SpCas9 variants more rewarding.

## Methods

**Materials**. Restriction enzymes, T4 ligase, Dulbecco's modified Eagle medium (DMEM), foetal bovine serum, Turbofect and penicillin/streptomycin were purchased from Thermo Fischer Scientific Inc. DNA oligonucleotides and the Gen-Elute HP Plasmid Miniprep and Midiprep kit used for plasmid purifications were acquired from Sigma-Aldrich. Q5 High-Fidelity DNA Polymerase, NEB5-alpha competent cells and HiFi Assembly Master Mix were purchased from New England Biolabs Inc.

**Plasmid construction**. Vectors were constructed using standard molecular biology techniques. All base editor coding sequences were cloned into the same plasmid backbone. For detailed cloning, primer and sequence information see

Supplementary Methods and Supplementary Table 2. The sequences of all plasmid constructs were confirmed by Sanger sequencing (Microsynth AG).

Plasmids acquired from the non-profit plasmid distribution service Addgene were the following: pU6-pegRNA-GG-acceptor (#132777[62]), pLenti-ABERA-P2A-Puro (#112675[22]), pLenti-FNLS-P2A-Puro (#110841[22]), pX330-Flag-wtSpCas9 (#92153[35]), pX330-Flag-dSpCas9 (#92113[35]), pX330-Flag-wtSpCas9-D10A (#80448), pcDNA3.1-mCherry (#128744), pCytERM_mScarlet_N1 (#85066[63]), pSc1-puro (#80438[64]), pmCherry-gRNA (#80457[35]), pX330-Flag-eSpCas9 (#126754), pX330-Flag-SpCas9-HF1 (#126755), pX330-Flag-HypaSpCas9 (#126756), pX330-Flag-Hypa-A-SpCas9 (#126757), pX330-Flag-evoSpCas9 (#126758), pX330-Flag-HeFSpCas9 (#126759)[38].

The following plasmids developed in this study are available from Addgene: pAT9624-BEAR-cloning (#162986), pAT9658-sgRNA-mCherry (#162987), pAT9679-sgRNA-BFP (#162988), pAT9651-BEAR-GFP (#162989), pAT9750-BEAR-mCherry (#162990), pAT9752-BEAR-mScarlet (#162991), pAT9650-BEAR-GFP-active (#162992), pAT9751-BEAR-mCherry-active (#162993), pAT9753-BEAR-mScarlet-active (#162994), pAT15415-BEAR-GFP-target-mCherry (#162995), pAT15416-BEAR-mScarlet-target-BFP (#162996), pAT9676-ABE (#162997), pAT9749-dABE (#162998), pAT9991-eABE (#162999), pAT9992-HF-ABE (#163000), pAT9993-Hypa-ABE (#163001), pAT9994-HypaR661A-ABE (#163002), pAT9995-evoABE (#163003), pAT9996-HeF-ABE (#163004), pAT9675-CBE (#163007), pAT9748-dCBE (#163008), pAT15064-eCBE (#163009), pAT15065-HF-CBE (#163010), pAT15066-Hypa-CBE (#163011), pAT15067-HypaA-CBE (#163012), pAT15068-evoCBE (#163013), pAT15069-HeF-CBE (#163014), pAT15516-BEAR-GFP-2in1 (#174082), pAT15482_ABE8e (#174120), pAT15483_e-ABE8e (#174121), pAT15484_HF-ABE8e (#174122), pAT15485_Hypa-ABE8e (#174123), pAT15486_HypaR661A-ABE8e (#174124), pAT15487_evo-ABE8e (#174125), pAT15488_HeF-ABE8e (#174126), pAT15489_dABE8e (#174127).

**Cell culturing and transfection**. N2a (neuro-2a mouse neuroblastoma cells, ATCC, CCL-131) and HEK293T (ATCC, CRL-1573) cells were grown at 37 °C in a humidified atmosphere of 5% $CO_2$ in DMEM medium supplemented with 10% heat-inactivated foetal bovine serum with 100 units/mL penicillin and 100 μg/mL streptomycin.

N2a and HEK293T cells cultured on 48-well plates were seeded a day before transfection at a density of $3 \times 10^4$ and $5 \times 10^4$ cells/well, respectively. 250 ng total DNA: 66 ng of BEAR target plasmid, 56 ng of sgRNA-mCherry (or sgRNA-BFP in case of BEAR-mScarlet) and 127 ng of CBE or 124 ng of ABE coding plasmid were mixed with 1 μL Turbofect reagent in 50 μL serum-free DMEM and were incubated for 30 minutes prior to being added to the cells. Three parallel transfections were made from each sample. Cells were analysed by flow cytometry three days after transfection. Flow cytometry gating strategy for these experiments is shown in Supplementary Fig. 11.

**Preparation of BEAR stable cell lines**. BEAR stable cell lines were prepared by modifying the self-cleaving plasmid method described earlier[64]. Inactive BEAR-mScarlet and BEAR-GFP constructs were cloned into a plasmid which bears a puromycin expression cassette and a Cas9 target that was used for self-cleaving plasmids (pSc-BEAR-mScarlet, pSc-BEAR-GFP). A corresponding spacer (which targets the pSc-BEAR plasmids) and AAVS1 genomic targets (described in ref. 65) were cloned into a sgRNA-mCherry plasmid. When these three and an SpCas9 coding plasmid are co-transfected into a cell, the sgRNA linearises the pSc-BEAR plasmid and integrates into the targeted locus (AAVS1) via non-homologous end joining.

HEK293T cells cultured on 6-well plates were seeded a day before transfection at a density of $5 \times 10^5$ cells/well. 1100 ng of pSc-BEAR, 800 ng of pSc-gRNA-mCherry, 800 ng of AAVS1-gRNA-mCherry and 1300 ng of SpCas9-HF1-plus[38] were mixed with 6 μL Turbofect in 400 μL serum-free DMEM and were incubated for 30 minutes prior to being added to the cells. Two days after transfection the cells were treated with 1 μg/μL puromycin for 15 days, then single cells were cloned in 96-well plates. Clones were checked for mCherry fluorescence negativity via flow cytometry and for Cas9 negativity via PCR analysis.

Stable cell lines were transfected with 250 ng total DNA: 76 ng sgRNA-mCherry (or sgRNA-BFP) and 174 ng of ABE coding plasmid were mixed with 1 μL Turbofect reagent in 50 μL serum-free DMEM and were incubated for 30 minutes prior to being added to the cells. Flow cytometry gating strategy for these experiments is shown in Supplementary Fig. 11.

**Flow cytometry and cell sorting**. Flow cytometry analysis was carried out using an Attune NxT Acoustic Focusing Cytometer (Applied Biosystems by Life Technologies). In all experiments, a minimum of 10,000 viable single cells were acquired by gating based on side and forward light-scatter parameters. BFP, GFP, mCherry and mScarlet signals were detected using the 405, 488, 561 and 561 nm diode laser for excitation and the 440/50, 530/30, 620/15 and 585/16 nm filter for emission, respectively. For data analysis Attune Cytometric Software v.2.1.0 was used.

In enrichment experiments, where fluorescence activated cell sorting was used, HEK293T cells, cultured on 48 well plates, were seeded a day before transfection at a density of $5 \times 10^4$ cells/well. 250 ng total DNA: 73 ng BEAR-GFP-2in1, 53 ng

genome targeting sgRNA-BFP, and 122 ng base editor coding plasmid were mixed with 1 μL Turbofect reagent in 50 μL serum-free DMEM and were incubated for 30 minutes prior to being added to the cells. In each parallel condition a total of 9 wells were transfected. Three days later, the cells were trypsinised and sorted directly into genomic lysis buffer, then genomic DNA was extracted. Three parallel transfections were made for each condition. Cell sorting was carried out on a FACSAria III cell sorter (BD Biosciences). The live single cell fraction was acquired by gating based on side and forward light-scatter parameters. BFP or GFP signals were detected using the 407 or 488 nm diode laser for excitation and the 450/50 or 530/30 nm filter for emission, respectively. To sort control (no enrichment) cells, live single cells were sorted regardless of any fluorescent markers. To sort transfection marker enriched cells, BFP positive cells were sorted regardless of GFP fluorescence. To sort BEAR enriched cells GFP positive cells, were sorted regardless of BFP fluorescence. A minimum of 50,000 cells were sorted in all experiments.

**Genomic DNA purification and EditR analysis.** Genomic DNA from FACS or other experiments was extracted according to the Puregene DNA Purification protocol (Gentra Systems Inc.). The purified genomic DNA was executed to PCR analysis, conducted with Q5 polymerase and locus specific primers (see Supplementary Table 2 and 3). PCR products were gel purified via NucleoSpin Gel and PCR Clean-up kit (Macherey-Nagel) and were Sanger sequenced. Editing efficiencies (on Fig. 3c,d) were analysed by EditR 1.0.9 (https://moriaritylab.shinyapps.io/editr_v10/)[43].

**Next-generation sequencing, indel and base editing frequency analysis.** In enrichment experiments base editing efficiency and indel frequency was analysed by NGS (Fig. 4). Amplicons for deep sequencing were generated using two rounds of PCR by Q5 high-fidelity polymerase to attach Illumina handles. The 1st step PCR primers used to amplify target genomic sequences and indexing of samples are listed in Supplementary Table 3. After the 2nd step PCR, samples were quantified using Qubit dsDNA HS Assay kit and PCR products were pooled for deep sequencing. Sequencing on an Illumina NextSeq instrument was performed by DeltaBio2000 Ltd. Reads were aligned to the reference sequence using BBMap.

Indels were counted computationally among the aligned reads that matched at least 75% to the first 20 bp of the reference amplicon. Indels without mismatches were searched at ±2 bp around the cut site with allowing indels of any size. For each sample, indel frequency was determined as (number of reads with an indel)/(number of total reads).

For each sample, base editing frequency was determined as the percentage of all sequencing reads with a target adenine converted to guanine (in the case of ABEs) or as the percentage of sequencing reads with a target cytosine converted to thymine (in the case of CBEs). For NGS analysis the following software were used: BBMap 38.08, samtools 1.8, BioPython 1.71, PySam 0.13. To avoid falsely high specificity ratios on Figs. 4e and 4h, during calculations indels lower than 0.05% were assumed to be 0.05% as this amount is considered to be the resolution limit of NGS. The deep sequencing data have been submitted to the NCBI Sequence Read Archive under accession number PRJNA748771.

**Statistics.** Unless stated otherwise, differences between samples were tested using Welch's one-way Anova with Games–Howell post hoc tests for samples with unequal variances and/or sample size and by one-way Anova with Tukey's post-hoc test for homoscedastic samples. Homogeneity of variances was tested by Levene's test. Statistical tests were performed using SPSS Statistics v.20 (IBM).

**Reporting summary.** Further information on research design is available in the Nature Research Reporting Summary linked to this article.

## Data availability
Expression vectors developed in this study are available from Addgene: pAT9624-BEAR-cloning (#162986), pAT9658-sgRNA-mCherry (#162987), pAT9679-sgRNA-BFP (#162988), pAT9651-BEAR-GFP (#162989), pAT9750-BEAR-mCherry (#162990), pAT9752-BEAR-mScarlet (#162991), pAT9650-BEAR-GFP-active (#162992), pAT9751-BEAR-mCherry-active (#162993), pAT9753-BEAR-mScarlet-active (#162994), pAT15415-BEAR-GFP-target-mCherry (#162995), pAT15416-BEAR-mScarlet-target-BFP (#162996), pAT9676-ABE (#162997), pAT9749-dABE (#162998), pAT9991-eABE (#162999), pAT9992-HF-ABE (#163000), pAT9993-Hypa-ABE (#163001), pAT9994-HypaR661A-ABE (#163002), pAT9995-evoABE (#163003), pAT9996-HeF-ABE (#163004), pAT9675-CBE (#163007), pAT9748-dCBE (#163008), pAT15064-eCBE (#163009), pAT15065-HF-CBE (#163010), pAT15066-Hypa-CBE (#163011), pAT15067-HypaA-CBE (#163012), pAT15068-evoCBE (#163013), pAT15069-HeF-CBE (#163014), pAT15516-BEAR-GFP-2in1 (#174082), pAT15482_ABE8e (#174120), pAT15483_e-ABE8e (#174121), pAT15484_HF-ABE8e (#174122), pAT15485_Hypa-ABE8e (#174123), pAT15486_HypaR661A-ABE8e (#174124), pAT15487_evo-ABE8e (#174125), pAT15488_HeF-ABE8e (#174126), pAT15489_dABE8e (#174127). Source data are provided with this paper. The deep sequencing data have been submitted to the NCBI Sequence Read Archive under accession number PRJNA748771. Source data are provided with this paper.

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

## Acknowledgements

We thank Dr. György Váradi and the FACS core facility for their valuable help with cell sorting. We thank Vivien Karl, Ildikó Szűcsné Pulinka, Judit Szűcs, Lilla Burkus and Judit Kálmán for their excellent technical assistance. We thank Antal Nyeste and Zsuzsa Bartos for their invaluable help and for Vanessza Laura Végi for proofreading the manuscript. The project was supported by grants K128188 and K134968 to E.W and PD134858 to P.I.K. from the Hungarian Scientific Research Fund (OTKA) and P.I.K. by 2018-1.1.1-MKI-2018-00167 and by ÚNKP-20-5-SE-20, and D.S. by VEKOP-2.1.7-15-2016-00393 from the National Research, Development and Innovation Office. P. I. K. is a recipient of the János Bolyai Research Scholarship of the Hungarian Academy of Sciences (BO/764/20). S.L.K was supported by grant EFOP-3.6.3-VEKOP-16-2017-00009 from the Higher Education Institutional Excellence Program of the Semmelweis University.

## Author contributions

A.T. conceived the idea of BEAR, designed and performed the experiments and interpreted the results. D.A.S. contributed to molecular cloning and to mismatch screens. P.I.K. contributed to creating stable BEAR cell lines. É.V. contributed to molecular cloning. S.L.K. analysed NGS data. E.W. designed the experiments, interpreted the results and supervised the research. A.T. and E.W. wrote the manuscript with input from all authors.

## Competing interests

The authors declare no competing interests.
