## [Peer Review File · Nature Communications]

Reviewers' Comments:

Reviewer #1:

Remarks to the Author:

Here, the authors present Base Editor Activity Reporter (BEAR), a new tool for quantifying base editor activity using a splicing-based fluorescent reporter approach. They show that BEAR functions in both plasmid and genomic contexts, and for both cytosine and adenine base editors. They use plasmid BEAR to enrich cells with base edits at arbitrary genomic loci, although much of the effect seems to be from selecting for transfected cells. They then use BEAR to analyze the effect of many SpCas9 enhanced fidelity variants both on base editor activity with both exact matching and mismatched sgRNA sequences. From these experiments, they conclude that for adenine base editors, enhanced fidelity SpCas9 variants (especially Hypa) modestly reduce editing mediated by exact matching sgRNAs but massively reduce editing mediated by mismatching sgRNAs. However, for cytosine base editors enhanced fidelity variants did not reduce editing mediated by mismatching sgRNAs relative to exact matching ones. A model is proposed to explain these differences, relating to the ability of cytosine, but not adenine, base editors to act on transiently separated DNA strands. The experiments appear to be well executed, with adequate replication. However, I have major reservations about the data analysis. I also think the authors don't do enough to explain how their system and its conclusions are novel/important since many other base editing reporters exist and many others have tested the effect of enhanced fidelity SpCas9 variants on base editing. Finally, the main point of the paper, which as far as I can tell is the model that accounts for differences in ABE/CBE specificity, needs more support.

Major points

-The title, abstract and introduction do not do a good job of explaining what this paper is actually about. Much of this is simple omission of information, but some of it is actually inaccurate description of what was done. For example, the abstract states "We have increased the efficiency of dead base editors to a level comparable to that of nickase base editors by enriching cells labelled for efficient base editing using Base Editor Activity Reporter (BEAR), a plasmid-based, fluorescent tool." To me, that suggests that the authors have somehow modified dead base editors to make them more efficient, which was not done.

-Related to the above point, the authors need to do a better job of setting out what has been done before (in clear, quantitative terms) and what is new in this manuscript. What, exactly, makes BEAR innovative? How is it different from existing methods? One of the main claims of the first half of the paper is that BEAR could be useful for high efficiency base editing without nickase base editors. Is that a problem that really needs solving? The authors need to make a convincing case with cites/data (e.g. specifics, not just vague claims).

-About half of the paper is occupied with using BEAR to test the on and off target activities of a large number of enhanced specificity SpCas9 base editor variants with different exact matching or mismatching sgRNAs. The main conclusion is that for CBEs, these enhanced specificity variants do not increase specificity as they do for ABEs. In the discussion, the authors posit that this is because CBEs may have more activity on targets where the separation of PAM-distal (e.g. edited) DNA strands is transient (e.g. like off targets). It would greatly enhance the paper to offer some direct experimental support of this idea, which would represent an important advance in our understanding of the mechanisms driving base editor off target effects. The recently developed ABE8e, which does not require full strand separation, would provide a facile means to test this hypothesis in the context of BEAR, and I suggest the authors do so.

-Related to the above point, the model proposed by the authors would seem to suggest a way to engineer more specific CBEs: reduce APOBEC activity. Has this been tried? If not, the authors might speculate about the possibilities.

-Overall, the manuscript needs editing for grammar and clarity. I mention some examples below, but my list is not exhaustive.

-I have major problems with the way the flow cytometry data is presented. If I understand the

figures/methods correctly, BEAR is always read out by flow cytometry (except where Sanger sequencing is used as confirmation). The authors typically report “% of fluorescent cells,” or a statistic derived from that metric. However, they never say how they decided what counts as a fluorescent cell. They should show example flow plots from each major experiment in supplemental figures, with gates drawn. They should also report the number of cells assessed and, especially, the number of cells in each gate (it’s fine if this is an amendment to their generic “a minimum of 10,000 viable, single cells were acquired...” statement in the methods).

-Related to the above major point, I wonder if by using a thresholding analysis approach, quantifying only the % of cells above a certain fluorescence level, the authors have lost something. For the plasmid version of BEAR, I imagine they might see graded as well as threshold effects because each cell contains hundreds or thousands of BEAR plasmids and might exhibit a graded response to increased editing.

Minor points

Page 2, line 11 - This is a small point, but base editors are designed to edit bases, regardless of whether they occur in genes or outside them.

Page 2, lines 17-22 - This last sentence of the abstract is long, dense and confusing. Consider rewriting.

Page 3, lines 48-50 - It is unclear from the text whether these existing markers have only been used with nCas9 so far, but could be used with dCas9, or whether they cannot be used with dCas9 at all. This should be unpacked/clearly explained so the reader understands the innovation presented here.

Page 3, line 62 - Here and elsewhere “increased fidelity variants,” are mentioned but not introduced. It seems like a brief mention of these, with citations and enough information to understand them, is important.

Pages 3 and 4, lines 64-67 - Here, “features and benefits” of base editor variants are mentioned along with “simple and effective means to compare performance.” These are vague references to the metrics we would actually want to compare the variants. The authors should strictly define these criteria for the reader (e.g. on-target efficacy, editing window, off-target, etc).

Page 3, lines 76-78 (and also the abstract) - I think the authors should explain more in both places about what BEAR is and how it improves upon existing methods (with actual quantitative metrics rather than vague claims).

Page 5, line 95 - 5’ss presumably means the 5’ splice site. If the authors are going to use this abbreviation they should define it. But I would just avoid abbreviating it at all.

Page 5 - The figures are cited out of order, which is confusing.

Page 5 and 6, Figure 2 - It takes quite a bit of switching back and forth between the text and the figure to really determine what is going on (e.g. what is “pre-edited,” why is plasmid 9 important relative to BEAR, etc). The information is all there, but it could be improved greatly in terms of readability.

Pages 5 and 6 - Because the details of the other base edit detection systems are not described in detail, the naive reader is going to have trouble judging whether what the authors are presenting here is actually better/different.

Page 7, line 155 - Rather than saying EditR was used, it would be better to say that Sanger sequencing was used to quantify editing with the EditR software. That way, readers who don’t know what EditR is will still understand the experiment.

Page 7, lines 153-156 - The ABE mismatched sgRNAs produced the expected graded diminution in

editing, but the CBE mismatches did not. The authors should at least comment on this oddity. Relatedly, it seems a little weird to report correlations for the CBE, since there are 3 tightly clustered points representing maximum signal and then one point with minimal signal (thus guaranteeing a high correlation). Instead, it might be better to just say "all sgRNAs except the three-base mismatch gave maximum signal in both contexts."

Page 7, lines 173-175 - I am confused that "20% of cells" are mScarlet positive but also "Thirty-one percent of cells in the transfected population exhibited mScarlet fluorescence." The authors should clarify. I also don't understand what "BEAR-enrichment," but I presume it is the idea of transfecting one BEAR color into a cell line with a second BEAR color in the genome and observing higher total editing rates. If that's so, I don't understand why the authors are presenting this data. It's totally unsurprising and doesn't add anything to the story at all. I think they are trying to set up the idea that the plasmid BEAR reporter can be used to enrich genomic edits, but if so this is very poorly explained.

Page 8, 181-190 - From the figure, it is clear that a transfection enrichment control was done and that it accounts for about 30-60% of the observed with BEAR enrichment. The text should make this clear, and I would argue that the fold-enrichment presented should be relative to the transfection enrichment control rather than no enrichment.

Page 8, 181-190 - Related to the above comment, it appears that the BEAR enrichment cells were selected both using the BEAR reporter (GFP) and the transfection reporter (mCherry). What does BEAR enrichment look like when ONLY the BEAR reporter (e.g. GFP) is used? This should be reported in the text, at least.

Page 8, "These experiments have indicated that BEAR facilitates base editing on genomic targets by dABE and dCBE without intentionally nicking the DNA, with efficiencies equal to or greater than that of nABE and nCBE." See one of my major comments, but I actually disagree with this framing of the result. The authors have not changed the efficiency of dABE/dCBE base editing. Instead, they have successfully recovered edited cells from the (unchanged) population of edited and unedited ones. Here and elsewhere, the authors should frame their claims more carefully.

Page 8, line 198 - "...only a few attempts of combining an increased fidelity variant with a base editor are reported in the literature." This statement is supported by six cites, which is a lot! The authors must do a better job of (briefly) justifying why more experiments are needed/interesting to probe enhanced fidelity variants in a base editor context.

Page 8, lines 218-221 - Both nABE and dABE are influenced by TadA deaminase and SpCas9, and nABE is additionally influenced by nicking activity. As written, the text does not reflect this reality.

Pages 8/9 - "These findings support that the activity profiles of increased fidelity ABE variants are primarily determined by the sequence specificities of the partner SpCas9 variants." I think this is an oversimplification. The sequence specificity of each of these variants is a consequence of the (variant) SpCas9*sgRNA complex, not just the variant itself.

Page 10 - The authors compare the correlations between editing efficiencies across all targets for the various ABE/CBE variants. They conclude "These data suggest that in strong contrast to ABEs, the activity profiles of the CBE variants are more determined by factors other than the properties of the increased fidelity SpCas9 nucleases, presumably by the sequence specificity of the deaminase partner and the subsequent repair process." I think this conclusion is way too strong and, possibly, unwarranted. The differences in correlation are not so large, and correlation across 20-30 targets is a poor metric anyhow. Could the authors do a deeper target-by-target analysis to better support this conclusion? At the very minimum, the statements here and on page 9 regarding the conclusions drawn from these analyses should be softened, and the correlation plots themselves should be shown in a supplementary figure.

Figure 3c, d - The authors should add the fit and correlation to the plots. Also, the meaning of the different shaped points (presumably the different mismatches) should be shown on the plot and/or mentioned in the text.

Reviewer #2:

Remarks to the Author:

In this manuscript, Ervin Welker et al. built a plasmid-based, fluorescent tool - BEAR to enrich successfully edited cells induced by dead base editors and examine the mismatch-tolerance of increased fidelity variants of base editors. This study revealed that, comparing to CBE, the Cas9-dependent off-target propensity of ABE editing can be effectively diminished by the application of increased fidelity variants and provided that BEAR can be used for comparing base editors with various features to select the best one for a particular task. Several points are suggested to revise this manuscript.

Major points

- 1) This study claimed that BEAR increases the efficiency of dCas9-derived base editors (dBE) to a level comparable to that of nCas9-derived base editors (nBE) in Fig.4. Strictly, the editing efficiency induced by dBE has not changed and the conclusion is inaccurate when compared nBE editing efficiency (without enrichment) with dBE (with BEAR enrichment).
- 2) The data shown in Fig. 2a and Fig. 2b are confusing. Fig. 2a-P9 and Fig. 2b-P9 have the same name, why? And in Fig. 2b, what is the difference between P23 and P9?
- 3) Fig. 1 demonstrated that BEARs with two kinds of non-canonical 5' splice sites (with Slightly different splicing efficiency shown in Supplementary Fig. 1) were used in ABE and CBE, so did the data shown in Fig. 6 exclude this factor? In addition, next-generation seq can make the conclusion more solid.

Minor points

- 1) The label in Fig. 4 does not match the text of the manuscript.
- 2) "To monitor transfection enrichment, fluorescence positive cell count was determined among the BFP positive cells which were the reporter for transfection efficiency." The source of the fluorescent signal of BFP should be indicated in the manuscript.

Reviewer #1 (Remarks to the Author):

Here, the authors present Base Editor Activity Reporter (BEAR), a new tool for quantifying base editor activity using a splicing-based fluorescent reporter approach. They show that BEAR functions in both plasmid and genomic contexts, and for both cytosine and adenine base editors. They use plasmid BEAR to enrich cells with base edits at arbitrary genomic loci, although much of the effect seems to be from selecting for transfected cells. They then use BEAR to analyze the effect of many SpCas9 enhanced fidelity variants both on base editor activity with both exact matching and mismatched sgRNA sequences. From these experiments, they conclude that for adenine base editors, enhanced fidelity SpCas9 variants (especially Hypa) modestly reduce editing mediated by exact matching sgRNAs but massively reduce editing mediated by mismatching sgRNAs. However, for cytosine base editors enhanced fidelity variants did not reduce editing mediated by mismatching sgRNAs relative to exact matching ones. A model is proposed to explain these differences, relating to the ability of cytosine, but not adenine, base editors to act on transiently separated DNA strands. The experiments appear to be well executed, with adequate replication. However, I have major reservations about the data analysis. I also think the authors don't do enough to explain how their system and its conclusions are novel/important since many other base editing reporters exist and many others have tested the effect of enhanced fidelity SpCas9 variants on base editing. Finally, the main point of the paper, which as far as I can tell is the model that accounts for differences in ABE/CBE specificity, needs more support.

Major points

1-The title, abstract and introduction do not do a good job of explaining what this paper is actually about. Much of this is simple omission of information, but some of it is actually inaccurate description of what was done. For example, the abstract states; "We have increased the efficiency of dead base editors editing to a level comparable to that of nickase base editors by enriching cells labelled for efficient base editing using Base Editor Activity Reporter (BEAR), a plasmid-based, fluorescent tool.;" To me, that suggests that the authors have somehow modified dead base editors to make them more efficient, which was not done.

We revised the abstract and the introduction to better emphasize the major points of the study and corrected the inaccurate statement:

"Using BEAR-enrichment, we increased the yield of base editing performed by nuclease inactive base editors to the level of the nickase versions while maintaining significantly lower indel background." (page 2, lines 4-6)

2-Related to the above point, the authors need to do a better job of setting out what has been done before (in clear, quantitative terms) and what is new in this manuscript. What, exactly, makes BEAR innovative? How is it different from existing methods? One of the main claims of the first half of the paper is that BEAR could be useful for high efficiency base editing without nickase base editors. Is that a problem that really needs solving? The authors need to make a convincing case with cites/data (e.g. specifics, not just vague claims).

In response to this point we prepared a table (**Supplementary Table 1**) that compares the existing methods to BEAR from several points of view, in quantitative terms. The table is presented here as well:

Method	Base editor used				Is the reporter transient?	Background signal	Number of targets tested	Number of possible targets	Can CBE and ABE be compared on the same target sequences?	Is enrichment with transient reporter demonstrated?	Can indels generate signal?
	nCBE	nABE	dCBE	dABE							
BE-FLARE	yes	no	no	no	yes	<0.5 %	1	1	no	yes (FACS)	ND
ACE	yes	no	no	no	yes	0-6%	1	1	no	no*	8% background signal with nCas9, 70% with Cas9
GFP panel	yes	no	no	no	yes	<0.5 %	3	restricted	no	no	ND
TREE	yes	no	no	no	yes	<0.5 %	1	1	no	yes (FACS)	ND
GO	yes	yes	no	no	ND	<0.5 %	10	restricted	no	no*	Not sensitive
BEON	no	yes	no	no	yes	5-20%	14	minimally restricted	no	yes (FACS)	ND
BEAR (this study)	yes	yes	yes	yes	yes	<0.5 %	79	minimally restricted	yes	yes (FACS)	Not sensitive

Supplementary Table 1 – Comparison of seven fluorescence-based markers of base editing

Seven fluorescent markers of base editing are compared (including this work). In the first column we compare whether the assay was demonstrated on detecting CBE or ABE using a nickase (nCBE, nABE) or nuclease inactive (dCBE, dABE) Cas9 partner. In the next column we compare if the method was used with a transient (plasmid) reporter. Next, we evaluate the amount of background fluorescent signal that is produced by the markers in a negative control condition (fluorescent marker alone without base editors). Next, the actual number of tested and the theoretically possible target sites are counted. Next, we compare whether the assay has the possibility of testing CBE and ABE on the exact same spacer sequences. After that, we indicate whether any base editing enrichment was demonstrated and if so, then whether it was with a transient or with a genomically integrated reporter. In the last column we reveal if any experiments were shown to detect the amount of generated fluorescent signal when the reporter is targeted by a nickase or a nuclease Cas9. “ND” abbreviation in the table means that that specific feature was not demonstrated in the publication. “*” abbreviation in the table means that enrichment was demonstrated in the publication but only on a genomically integrated reporter, not on a transient one.

Improving the specificity of editing is a continuous effort in all aspects of possible off-targets/byproducts, whether it is SpCas9-dependent or independent, proximity or genome wide, and whether off-target editing occurs on either DNA or RNA. While considerable effort has been made to decrease the bystander indel formation of CBEs, its level is still generally larger with a magnitude than that of ABEs. The amount of bystander indel formation is sequence-context dependent and there are several reports of it considerably exceeding^{1, 2, 3, 4, 5} the about 1-2% that is expected to occur. In the revised manuscript, when we used nCBE to edit target cytosines on 5 different loci it resulted in between 2-15% indel formation. Using BEAR to enrich dCBE edited cells we have substantially lowered the amount of bystander indels (0.07%-4% with dCBE and under the detection limit of conventional NGS with dABE). It improved the specificity (base editing/indel ratio) by 3.1- to 30.1-fold compared to nCBE, and by 1.1- to 21.9-fold compared to nABE while enriching base edited cells to approach the level of nCBE and nABE, respectively. The results are presented in the revised **Figure 4**. The figure is presented here as well:

Figure 4 – Enrichment of dABE and dCBE base edited cells with BEAR, matches the efficiency of nABE and nCBE while creating less indels

(a) Either the BEAR-mScarlet or the BEAR-GFP plasmid and dABE were co-transfected into the BEAR-GFP or BEAR-mScarlet cell line and the cells were analysed by flow cytometry gated for GFP or mScarlet positive cells, respectively. Cells without additional gating: no enrichment (black bars) with additional BFP gate: transfection enrichment (grey bars), with additional BEAR-plasmid colour gate: BEAR enrichment (purple bars). To monitor transfection efficiency a blue fluorescent protein (BFP) was placed onto the sgRNA-expressing plasmid.

(b) The target sequences of the endogenous genomic targets edited with CBE or ABE variants in panels c and f. Bases edited most efficiently are coloured and their efficiencies are depicted on panels c and f. For all other edited bases and editing efficiencies see Supplementary Data.

The BEAR-GFP-2in1 plasmid and endogenous genomic targets were co-edited by dCBE (c-e) and dABE (f-h) and analysed by NGS. Edited cells were sorted to 3 fractions: all cells (no enrichment, black), BFP positive cells (transfection enrichment, grey), and cells with GFP positivity representing base editing enriched cells (dCBE – light green, dABE – light purple). In all experiments nCBE (dark green) and nABE (dark purple) edited cells were monitored without enrichment as controls. Base editing and indel formation was quantified from the same samples in the case of dCBE (c,d) and dABE (f,g) edited cells. The specificity (base editing % / indel %) is displayed for nCBE and dCBE (e) as well as for nABE and dABE (h). Columns represent means +/- SD of three parallel transfections. When indel % was lower than the detection limit of NGS (dashed grey line, 0.05%), specificity was calculated with 0.05% indel to avoid falsely high specificity values. Differences between samples were tested using ANOVA. NS:p>0.05, *:p<0.05, **:p<0.01, ***:p<0.001. For source data see Supplementary Data.

3-About half of the paper is occupied with using BEAR to test the on and off target activities of a large number of enhanced specificity SpCas9 base editor variants with different exact matching or mismatching sgRNAs. The main conclusion is that for CBEs, these enhanced specificity variants do not increase specificity as they do for ABEs. In the discussion, the authors posit that this is because CBEs may have more activity on targets where the separation of PAM-distal (e.g. edited) DNA strands is transient (e.g. like off targets). It would greatly enhance the paper to offer some direct experimental support of this idea, which would represent an important advance in our understanding of the mechanisms driving base editor off target effects. The recently developed ABE8e, which does not require full strand separation, would provide a facile means to test this hypothesis in the context of BEAR, and I suggest the authors do so.

We fully agree with this suggestion of the Reviewer, that ABE8e provides a unique opportunity to test our hypothesis. We performed both on-target and off-target screens with ABE8e employing the same set of six increased-fidelity nuclease variants. If our hypothesis is correct, nABE8e with HeF- and evo-SpCas9 partners (similarly to CBE but contrasting ABE7), should show higher average on-target editing than its inactive nuclease variant (dABE8e). Furthermore, increased fidelity nucleases should not be very effective to increase its mismatch tolerance similarly to CBE but contrasting ABE7. It is exactly what we see: HeF-ABE8e and evo-ABE8e reached average on-target activity of 52% and 61%, respectively, exceeding that of dABE8e (31%). Furthermore, increased fidelity variant partners can decrease off-target cleavage only when the sgRNAs contain at least 3 mismatches, in contrast to ABE7 with increased fidelity Cas9 partners, which have almost zero activity already with two mismatches. These results are presented in **Figure 7** and discussed in page 13, lines 16-30 and in page 14 lines 1-18. The figure is presented here as well:

Figure 7 – On- and off-target activities of various ABE8e variants

(a) The heatmap shows mean on-target activity of three parallel transfections normalised to the active splice donor site plasmid. A set of 34 previously used targets (Fig. 5) were used to determine on-target editing efficiencies of dABE8e, nABE8e and 6 increased fidelity ABE8e base editors.

(b) Mismatch tolerance of ABE8e and its increased fidelity variants were compared utilising the exact same matching sgRNA (target 1 in Fig. 5) and 50 sgRNAs mismatching in one, two, three, four or five positions as indicated as red letters. Blue and yellow heatmaps show the mean normalised activity (off-target/on-target) derived from three parallel transfections. White and red heatmaps show the mean on-target activity (percentages of GFP positive cells) derived from three parallel transfections. For source data see Supplementary Data.

4-Related to the above point, the model proposed by the authors would seem to suggest a way to engineer more specific CBEs: reduce APOBEC activity. Has this been tried? If not, the authors might speculate about the possibilities.

We also realized it and speculated this possibility in the manuscript on page 17 lines 18-22. We did not have a chance to generate such new variant since the process would likely involve complex optimization of the mutations; they would need to reduce the deaminase activity of APOBEC but should still maintain it on a level, which ensures efficient base editing.

5-Overall, the manuscript needs editing for grammar and clarity. I mention some examples below, but my list is not exhaustive.

We have edited the manuscript for clarity.

6- I have major problems with the way the flow cytometry data is presented. If I understand the figures/methods correctly, BEAR is always read out by flow cytometry (except where Sanger sequencing is used as confirmation). The authors typically report of fluorescent cells or a statistic derived from that metric. However, they never say how they decided what counts as a fluorescent cell. They should show example flow plots from each major experiment in supplemental figures, with gates drawn. They should also report the number of cells assessed and, especially, the number of cells in each gate (its fine if this is an amendment to their generic minimum of 10,000 viable, single cells were acquired; statement in the methods).

We presented the plot requested by the Reviewer as **Supplementary Fig. 11** and showed how we determine the position of the gate using the negative control (no fluorescence) population, and the gating strategy for each major experiment. The figure is presented here as well:

Supplementary Figure 11 – Flow cytometry gating examples

Flow cytometry gating examples are shown for different BEAR experiments. On panel (a) a density plot of HEK293T cells is shown when all live single cells (determined by FSC and SSC parameters) are either (1) untransfected (left plot) and thus show no fluorescence for mCherry or GFP, (2) co-transfected with an sgRNA-mCherry plasmid and the BEAR-GFP plasmid but with a dCas9, thus displaying mCherry but no GFP fluorescence, (3) or sgRNA-mCherry and BEAR-GFP plasmid is co-transfected with an ABE expressing plasmid, causing the splice site to be edited, and thus displaying both mCherry and GFP fluorescence. On panel (b) a gating example is shown from Fig. 4a where cells harbouring genomically

integrated copies of BEAR-mScarlet sequences are co-edited alongside the BEAR-GFP plasmid. On the upper left panel mScarlet positive cells are counted in all cells regardless of the presence of another fluorescent protein. The lower left panel shows mScarlet positive cells gated in the BFP positive population which accounts for the transfection marker enriched cells. On the upper right panel GFP and BFP positive cells are gated, and in this population mScarlet positive cells are counted (BEAR-GFP enrichment). On panel (c) a gating example is shown from Fig. 4a where cells harbouring genomically integrated copies of BEAR-GFP sequences are co-edited alongside the BEAR-mScarlet plasmid. On the upper left panel GFP positive cells are counted in all cells regardless of the presence of another fluorescent protein. The lower left panel shows GFP positive cells gated in the BFP positive population which accounts for the transfection marker enriched cells. On the upper right panel mScarlet and BFP positive cells are gated, and in this population GFP positive cells are counted (BEAR-mScarlet enrichment).

Furthermore, we gave a statement in the methods part about the minimal number of single cells acquired in the transfection gate and in the GFP or mScarlet gate on page 20 lines 26-28 and on page 21 lines 1-2. Generally, our transfection efficiency in HEK293T cells is above 80%, meaning that there are at least 8000 cells in the transfection marker gate. In the other gates the actual numbers are dependent on the particular experiment. The measured percentages are significantly above the background when there are at least 200 cells in the GFP/mScarlet gate.

7-Related to the above major point, I wonder if by using a thresholding analysis approach, quantifying only the % of cells above a certain fluorescence level, the authors have lost something. For the plasmid version of BEAR, I imagine they might see graded as well as threshold effects because each cell contains hundreds or thousands of BEAR plasmids and might exhibit a graded response to increased editing.

This suggestion of the Reviewer is definitely plausible, we also considered both options for presenting the flow cytometric data. To decide which approach should be used when a great number of plasmids are present in the cell instead of the usual few genomic copies, we executed an experiment, where we evaluated both approaches by the fluorescence of the cells containing an increasing number of plasmids expressing edited (i.e., functional fluorescent) proteins. The results presented here show that fluorescence intensity provides gradual increase in the whole range examined, however, show low sensitivity for the lower ranges. The threshold, %-reporting approach provides data that are more sensitively reported in the lower ranges, however, reaches a plateau at about 80% fluorescence. Since our data are generally lower than 80%, approaching 80% in the on-target screen of ABE with a few targets, in order to provide a higher sensitivity reporting we decided to use the threshold %-reporting approach to present our flowcytometric data.

Different amounts (0-200 ng) of BEAR-GFP plasmid were co-transfected with ABE and the appropriate sgRNA. The number of GFP positive cells were determined by flow cytometry (red, plotted on the left y axis), and from the same samples the intensity of the GFP signal is depicted as the means of arbitrary fluorescence units (AU, grey, plotted on the right y axis). During transfection the total amount of plasmids

transfected was kept constant by using a mock plasmid (with the same size as BEAR-GFP). Columns represent means +/- SD of three parallel transfections.

Minor points

Page 2, line 11 - This is a small point, but base editors are designed to edit bases, regardless of whether they occur in genes or outside them.

Thanks for catching it. We corrected it.

Page 2, lines 17-22 - This last sentence of the abstract is long, dense and confusing. Consider rewriting.

We have revised it. Now it reads as:

“Furthermore, by exploiting the semi-high-throughput potential of BEAR, we examined whether increased fidelity SpCas9 variants could be used to decrease SpCas9-dependent off-target effects of ABE and CBE. Comparing them on the same target sets revealed that CBE remains active on sequences, where increased fidelity mutations and/or mismatches decrease the activity of ABE. Our results suggest that the deaminase domain of ABE is less effective to act on rather transiently separated target DNA strands, than that of CBE explaining its lower mismatch tolerance.”

Page 3, lines 48-50 – It is unclear from the text whether these existing markers have only been used with nCas9 so far, but could be used with dCas9, or whether they cannot be used with dCas9 at all. This should be unpacked/clearly explained so the reader understands the innovation presented here.

Reporting the action of dead base editors has not yet been reported for any other existing markers. We do not know which marker could be made compatible with the use of dead base editors by appropriate optimisation of the approach. This fact is indicated in Supplementary Table 1, that shows the features of these markers in relation to BEAR (see also at the reply to the second comment above) and stated in the manuscript on page 3 lines 9-12:

“Unfortunately, the markers that have been developed to date for the enrichment of CBE- or ABE-edited cells exclusively employ nickase SpCas9 leaving the question open, whether they are sensitive enough to report on the activity of dead base editors.”

Page 3, line 62 - Here and elsewhere increased fidelity variants are mentioned but not introduced. It seems like a brief mention of these, with citations and enough information to understand them, is important.

Now, we gave a short introduction and references for increased fidelity variants on page 3 lines 25-29 and on page 4 lines 1-2:

“One of the promising approaches is to apply increased fidelity variants of SpCas9^{30, 31, 32, 33, 34}. These variants have been introduced in order to decrease the off-target editing of the nuclease version of SpCas9. These variants exhibit higher specificity and decreased activity in a target-dependent manner, seemingly trading efficiency for specificity^{34, 35, 36, 37}. Scientific literature in this area is lacking a thorough assessment of the applicability of increased fidelity SpCas9 variants to decrease the mismatch tolerance of ABE and CBE. Thus, it has been designated as the second objective of our study.”

Pages 3 and 4, lines 64-67 - Here, features and benefits of base editor variants are mentioned along with simple and effective means to compare performance. These are vague references to the metrics we would actually want to compare the variants. The authors should strictly define these criteria for the reader (e.g. on-target efficacy, editing window, off-target, etc,

We revised the implicated sentences as follows:

“The absence of simple and effective means to compare the performance of these base editors, in terms of on-target efficiency, tolerance for mismatches and relative activity at different positions of the extended editing-window on various sequences and in any cell of choice, hampers the exploitation of base editor variants to their full potential.”

Page 3, lines 76-78 (and also the abstract) - I think the authors should explain more in both places about what BEAR is and how it improves upon existing methods (with actual quantitative metrics rather than vague claims).

On page 4 lines 24-28 in the revised manuscript we now describe BEAR as:

“BEAR, the assay we designed in accordance with these requirements, is based on a split GFP protein separated by the last intron of the mouse Vim gene. The sequence of the functional 5' splice site (hereafter referred to as splice donor site) is altered to abolish splicing and thus GFP fluorescence, but both splicing and GFP fluorescence can be restored by applying base editors (Fig. 1).”

We also compare BEAR with existing markers in Supplementary Table 1. (it is also shown at the reply to the second comment above).

Page 5, line 95 - 5'ss presumably means the 5' splice site. If the authors are going to use this abbreviation they should define it. But I would just avoid abbreviating it at all.

We no longer use the abbreviation and refer to the 5' splice site as splice donor site everywhere in the revised manuscript.

Page 5 - The figures are cited out of order, which is confusing.

We had difficulties finding a reasonable arrangement for the panels within the figure, especially that we could not put panel 'a' to the left-top corner of the figure. It felt less confusing to cite the panels out of order than having a panel 'a' somewhere in the middle of the figure. Now, the figure is revised, and we resolved this issue (see the figure in the next comment).

Page 5 and 6, Figure 2 - It takes quite a bit of switching back and forth between the text and the figure to really determine what is going on (e.g. what is pre-edited why is plasmid 9 important relative to BEAR, etc). The information is all there, but it could be improved greatly in terms of readability.

We revised Figure 2 (and Supplementary Figure 2 as well) and the text on page 5 lines 8-29 and on page 6 lines 1-20 for an easier read:

Figure 2 – Splice site variants for identifying candidate BEAR sequences

Flow cytometry measurements of GFP positive HEK293T cells, transfected with plasmids harbouring systematically altered splice sites (expected “inactive” sequences), which hold the possibility of being converted by ABE (a) or CBE (b) to sequences expected to be functional, “active” splice sites. BEAR plasmids with inactive and active splice sites were generated by molecular cloning, the latter representing the maximum fluorescence that can be achieved by base editing. The sequences between the column charts represent the intended “inactive” or “active” splice donor site and flanking sequence pairs. Letters highlighted in blue indicate the bases that correspond to the canonical 5' - G GT AAGT - 3' sequence (upper panels); the altered bases in the GT splice donor sites are underlined. Five sequence pairs (P1-P5) with minimal fluorescence for the inactive and maximal fluorescence for the active splice donor site were selected for further analyses as detailed in Fig. 2c-e. Note that the target sequence in the a and b panels are different. Flow cytometry measurements of GFP positive HEK293T (c and e) and N2a (d) cells co-transfected with a selected reporter plasmid harbouring an inactive splice site and base editor or control nuclease constructs as indicated in the figure. Columns represent means, +/- SD of three parallel transfections.

Pages 5 and 6 - Because the details of the other base edit detection systems are not described in detail, the naive reader is going to have trouble judging whether what the authors are presenting here is actually better/different.

This refers to main point 2 and some related minor points. Now, we created a table (Supp. Table 1. also shown at the response to the second comment above) to show how BEAR is different from other base editor reporting approaches.

Page 7, line 155 – Rather than saying EditR was used, it would be better to say that Sanger sequencing was used to quantify editing with the EditR software. That way, readers who don't know what EditR is will still understand the experiment.

Thanks, we corrected it on page 7 lines 13-17. Now it reads as:

“To confirm that fluorescence recovery is the result of successful base editing, we have employed a fully matching sgRNA, and sgRNAs with one, two or three mismatches with ABE (Fig. 3c) or CBE (Fig. 3d) on the BEAR-GFP cell line, and monitored base editing activity by measuring the number of GFP positive cells, as well as by using Sanger sequencing to quantify editing with the EditR software⁴².”

Page 7, lines 153-156 - The ABE mismatched sgRNAs produced the expected graded diminution in editing, but the CBE mismatches did not. The authors should at least comment on this oddity. Relatedly, it seems a little weird to report correlations for the CBE, since there are 3 tightly clustered points representing maximum signal and then one point with minimal signal (thus guaranteeing a high correlation). Instead, it might be better to just say all sgRNAs except the three-base mismatch gave maximum signal in both contexts.

We followed the Reviewer's suggestion and corrected the figure and the text (page 7 lines 17-20):

“The measured fluorescence intensity was found to be proportional to the level of actual base editing ($r=0.98$ in case of ABE). In the case of CBE, all sgRNAs except the one with a three-base mismatch gave maximum signal in both contexts. This different mismatch tolerance of ABE and CBE is examined in more detailed later in this study.”

Page 7, lines 173-175 - I am confused that 20% of cells are mScarlet positive but also Thirty-one percent of cells in the transfected population exhibited mScarlet fluorescence. The authors should clarify. I also don't understand what BEAR-enrichment but I presume it is the idea of transfecting one BEAR color into a cell line with a second BEAR color in the genome and observing higher total editing rates. If that's so, I don't understand why the authors are presenting this data. It's totally unsurprising and doesn't add anything to the story at all. I think they are trying to set up the idea that the plasmid BEAR reporter can be used to enrich genomic edits, but if so this is very poorly explained.

We revised the text at page 8, lines 6-9 to better explain this idea by inserting the following sentence:

“First, as a proof of principle, we used the BEAR plasmids in combination with genome-integrated copies of different BEAR colours to see if they indeed can label cells, in which base editing events had taken place at genome-integrated targets.”

Page 8, 181-190 - From the figure, it is clear that a transfection enrichment control was done and that it accounts for about 30-60% of the observed with BEAR enrichment. The text should make this clear, and I would argue that the fold-enrichment presented should relative to the transfection enrichment control rather than no enrichment.

We also considered that BEAR enrichments should clearly exceed transfection enrichments, this is the reason why we show transfection enrichments in all figures. However, we think that to present fold enrichments relative to no enrichment is more unambiguous. Transfection enrichment, i.e., the population, showing the presence of the transfection control, depends on many factors that are mainly independent from the real efficiency of the actual transfection. Such factors are the spectral propensity, the employed fluorescent protein, the actual sensitivity of the used flow-cytometer set up to the employed colour, the level of expression that the expression cassette of the fluorescent transfection marker protein can provide and the amount of transfection marker plasmid used in the experiments, just to mention a few. The figures clearly show the transfection enrichments under the used conditions in all experiments, so the readers can

make their own judgement. However, in the text we prefer to report fold BEAR-enrichments relative to no enrichments due to the above reasons.

Also, the yellow laser of our sorter (used for mCherry excitation) broke when doing the revisions and during the pandemic the service could not fix it. As a quick workaround we replaced the transfection marker of mCherry to BFP in all our constructs and excited the protein with a still working violet laser. This altered the transfection enrichments results on Figure 4c-h demonstrating how the transfection enrichments depends on these conditions. We have also updated the used laser/filter set up for cell sorting in the materials and methods section, accordingly.

Page 8, 181-190 - Related to the above comment, it appears that the BEAR enrichment cells were selected both using the BEAR reporter (GFP) and the transfection reporter (mCherry). What does BEAR enrichment look like when ONLY the BEAR reporter (e.g. GFP) is used? This should be reported in the text, at least.

The fluorescent cassette (BFP or mCherry) is always placed into the sgRNA-expressing plasmid for reporting transfection enrichments. There is no significant difference, whether the BEAR-enrichments are monitored in both using the BEAR reporter AND the transfection reporter or ONLY the BEAR reporter as demonstrated in the figure below:

The BEAR-GFP plasmid and dABE were co-transfected into the BEAR-mScarlet cell line, and the cells were analysed by flow cytometry. To monitor cells with no enrichment (no enrichment, black bars), mScarlet positive cell count was determined among all live single cells gated via flow cytometry. To monitor transfection enrichment (transfection enrichment, gray bars), mScarlet positive cell count was determined among the BFP positive cells which were the reporter for transfection efficiency. BEAR enriched cells were measured by counting the amount of mScarlet positive cells among the GFP positive cells (BEAR enrichment, purple bars). Enrichment was also monitored when the BEAR-mScarlet plasmid was used as the enrichment reporter in BEAR-GFP cell lines. BEAR enrichment is also depicted when GFP or mScarlet positive cells were not gated in the transfected population (pink). Differences between samples were tested using Welch's one-way Anova with Games-Howell post hoc test. Homogeneity of variances was tested by Levene's test. Columns represent means +/- SD of three parallel transfections. NS : $p > 0.05$, ** : $p < 0.01$, *** : $p < 0.001$,

However, in the revised manuscript following the Reviewer's advice, all genomic BEAR enrichments (Figure 4) were done as cells sorted for GFP fluorescence regardless of the transfection marker fluorescence. This is also stated in the materials and methods section on page 21 lines 6-9:

"To sort control (no enrichment) cells, live single cells were sorted regardless of any fluorescent markers. To sort transfection marker enriched cells, BFP positive cells were sorted regardless of GFP fluorescence. To sort BEAR enriched cells GFP positive cells,

were sorted regardless of BFP fluorescence. A minimum of 50,000 cells were sorted in all experiments.”

Page 8, These experiments have indicated that BEAR facilitates base editing on genomic targets by dABE and dCBE without intentionally nicking the DNA, with efficiencies equal to or greater than that of nABE and nCBE. See one of my major comments, but I actually disagree with this framing of the result. The authors have not changed the efficiency of dABE/dCBE base editing. Instead, they have successfully recovered edited cells from the (unchanged) population of edited and unedited ones. Here and elsewhere, the authors should frame their claims more carefully.

This is the only other suggestion we disagree with the comments of the Reviewer. While it was erroneous to mention “increasing the efficiency of dead base editors” (thanks for catching it in previous points) we think that it is correct to say that we improved the efficiency of base editing, since we altered the population of edited and unedited cells by enriching for the edited ones. In our point of view, the term of “base editing” refers **both** to the events occurring in the cells, by the action of the base editor on a particular target **and** to the genome modifying methods that may start with the design of the sgRNA sequence and ends with the verification of the successful mutation in individual cells or in a resulted population, from which one may generate the individual clones. BEAR enrichment clearly increases the yield of the second process. We revised the text (page 9 lines 16-18) to more precisely reflect that we refer to the latter meaning of base editing:

“Altogether, these experiments indicate that BEAR-enrichment yields base edited genomic targets with about nickase-level (nABE and nCBE) efficiency while preserving the low indel background of dABE and dCBE.”

Page 8, line198 - ...only a few attempts of combining an increased fidelity variant with a base editor are reported in the literature. This statement is supported by six cites, which is a lot! The authors must do a better job of (briefly) justifying why more experiments are needed/interesting to probe enhanced fidelity variants in a base editor context.

We revised the text on page 9 line 21-25 according to the reviewer suggestion:

“Applying increased fidelity variants may seem to be a plausible approach to decrease the Cas9-dependent off-target effects of base editors. However, no study provides a thorough assessment of this alternative, although a few attempts of combining an increased fidelity variant with a base editor are reported in the literature”.

Page 8, lines 218-221 – Both nABE and dABE are influenced by TadA deaminase and SpCas9, and nABE is additionally influenced by nicking activity. As written, the text does not reflect this reality.

As a matter of fact, we intended to write the same in the sentences:

“In theory, the activity profile of dABE is influenced by the sequence specificity of both the TadA deaminase and the binding of SpCas9. In contrast, the activity profile of nABE is also influenced by the nicking activity of SpCas9, which aims to bias the repair system in order to correct the mismatching bases of the unedited strand, and thus, to increase editing efficiency.”

To make it clearer we revised the text on page 10, lines 18-22 to:

“In theory, the activity profile of dABE is influenced by the sequence specificity of both the TadA deaminase and the binding of SpCas9. In contrast, the activity profile of nABE is additionally influenced by the nicking activity of SpCas9, which aims to bias the repair system

into correcting the mismatching bases of the unedited strand, thus to increase editing efficiency.”

Pages 8/9 – These findings support that the activity profiles of increased fidelity ABE variants are primarily determined by the sequence specificities of the partner SpCas9 variants. I think this is an oversimplification. The sequence specificity of each of these variants is a consequence of the (variant) SpCas9*sgRNA complex, not just the variant itself.

We also meant that, however, this sentence is deleted from the manuscript now.

Page 10 – The authors compare the correlations between editing efficiencies across all targets for the various ABE/CBE variants. They conclude These data suggest that in strong contrast to ABEs, the activity profiles of the CBE variants are more determined by factors other than the properties of the increased fidelity SpCas9 nucleases, presumably by the sequence specificity of the deaminase partner and the subsequent repair 262 process. I think this conclusion is way too strong and, possibly, unwarranted. The differences in correlation are not so large, and correlation across 20-30 targets is a poor metric anyhow. Could the authors do a deeper target-by-target analysis to better support this conclusion? At the very minimum, the statements here and on page 9 regarding the conclusions drawn from these analyses should be softened, and the correlation plots themselves should be shown in a supplementary figure.

We are convinced by the reviewer’s reasoning in this point and deleted this argument from the revised manuscript. Furthermore, at the corresponding points we report on the correlation we present as scatter plots as well on Supplementary Figures (Sup. Fig. 3e; Sup. Fig. 5; Sup. Fig. 6b,d;.Sup. Fig. 7d; Sup. Fig. 8; Sup. Fig. 10)

Figure 3c, d – The authors should add the fit and correlation to the plots. Also, the meaning of the different shaped points (presumably the different mismatches) should be shown on the plot and/or mentioned in the text.

This point is related to the comment on Page 7, lines 153-156. We revised the figure and the text accordingly.

We would like to thank to this Reviewer for their helpful comments and suggestions that helped us to improve the quality of our manuscript.

Reviewer #2 (Remarks to the Author):

In this manuscript, Ervin Welker et al. built a plasmid-based, fluorescent tool – BEAR to enrich successfully edited cells induced by dead base editors and examine the mismatch-tolerance of increased fidelity variants of base editors. This study revealed that, comparing to CBE, the Cas9-dependent off-target propensity of ABE editing can be effectively diminished by the application of increased fidelity variants and provided that BEAR can be used for comparing base editors with various features to select the best one for a particular task. Several points are suggested to revise this manuscript.

Major points

1) This study claimed that BEAR increases the efficiency of dCas9-derived base editors (dBE) to a level comparable to that of nCas9-derived base editors (nBE) in Fig.4. Strictly, the editing efficiency induced by dBE has not changed and the conclusion is inaccurate when compared nBE editing efficiency (without enrichment) with dBE (with BEAR enrichment).

This point refers to the comment of Reviewer 1 on page 14. We corrected the inaccurate sentences on page 9 lines 16-18:

“Altogether, these experiments indicate that BEAR-enrichment yields base edited genomic targets with about nickase-level (nABE and nCBE) efficiency while preserving the low indel background of dABE and dCBE.”

2) The data shown in Fig. 2a and Fig. 2b are confusing. Fig. 2a-P9 and Fig. 2b-P9 have the same name, why? And in Fig. 2b, what is the difference between P23 and P9?

Since the original Figure 2 apparently was not easy to follow, as indicated by both reviewers, we revised the figures so now we hope that they are not confusing anymore. We changed the names of the constructs in Fig 2a and Fig2b.

For ABE base editor and CBE base editor reporter candidates, two different sequences were used for the variable region of the target, one for 2a (target ABE: GCAGGTAAGTGCATAGACTG) and one for 2b (target CBE: GCAGGTAAGTGTCTGGAGGTG), which is now also shown in the revised figure. In the case of Fig2a P9 and Fig2b P9 they have the same name because the disrupted (now called inactive) plasmid with the target/splice site (AC) is the same in both panels. (However, the corresponding “pre-edited” -now called active; mimicking the successfully edited sequences for ABE: GC, in fig 2a, and for CBE: AT in fig 2b- plasmids are different.)

The splice site sequences are the same for P23 and for P9, however, the rest of the target sequences are different in the variable part, P9 has target1 and P23 has target2. To avoid all these confusions, we now reorganized the figures and changed the used terminology for the plasmids. We revised Supplementary Figure 2 as well.

The revised Figure 2 now looks like this:

Figure 2 – Splice site variants for identifying candidate BEAR sequences

Flow cytometry measurements of GFP positive HEK293T cells, transfected with plasmids harbouring systematically altered splice sites (expected “inactive” sequences), which hold the possibility of being converted by ABE (a) or CBE (b) to sequences expected to be functional, “active” splice sites. BEAR plasmids with inactive and active splice sites were generated by molecular cloning, the latter representing the maximum fluorescence that can be achieved by base editing. The sequences between the column charts represent the intended “inactive” or “active” splice donor site and flanking sequence pairs. Letters highlighted in blue indicate the bases that correspond to the canonical 5' - G GT AAGT - 3' sequence (upper panels); the altered bases in the GT splice donor sites are underlined. Five sequence pairs (P1-P5) with minimal fluorescence for the inactive and maximal fluorescence for the active splice donor site were selected for further analyses as detailed in Fig. 2c-e. Note that the target sequence in the a and b panels are different. Flow cytometry measurements of GFP positive HEK293T (c and e) and N2a (d) cells co-transfected with a selected reporter plasmid harbouring an inactive splice site and base editor or control nuclease constructs as indicated in the figure. Columns represent means, +/- SD of three parallel transfections.

3) Fig. 1 demonstrated that BEARs with two kinds of non-canonical 5'; splice sites (with Slightly different splicing efficiency shown in Supplementary Fig. 1) were used in ABE and CBE, so did the data shown in Fig. 6 exclude this factor? In addition, next-generation seq can make the conclusion more solid.

The data we gain using the base editors are normalized to the fluorescence of the corresponding pre-edited (now called active) plasmids to account for the effect of the slightly differing splicing efficiency. We checked all targets used here for the splicing efficiency of their corresponding pre-edited (active) plasmids, to exclude this factor in all experiments of the study.

We also applied NGS during the revision to make our conclusion more solid for the experiments shown in (Fig. 4) that are now shown and described in the revised version on page 8 lines 19-28 and on page 9 lines 1-18.

Minor points

1) The label in Fig. 4 does not match the text of the manuscript.

Thanks for catching it, we corrected it.

2) To monitor transfection enrichment, fluorescence positive cell count was determined among the BFP positive cells which were the reporter for transfection efficiency. The source of the fluorescent signal of BFP should be indicated in the manuscript.

The BFP is placed into the sgRNA-expressing plasmid, as indicated in the Materials and methods section on page 21 line 5 and in the legend of Fig. 4 in the revised manuscript:

“To monitor transfection efficiency a blue fluorescent protein (BFP) was placed to the sgRNA-expressing plasmid.

We would like to thank this Reviewer for their helpful comments and suggestions that helped us to improve the quality of our manuscript.

REFERENCES

1. Komor AC, Kim YB, Packer MS, Zuris JA, Liu DR. Programmable editing of a target base in genomic DNA without double-stranded DNA cleavage. *Nature* **533**, 420-424 (2016).
2. Komor AC, *et al.* Improved base excision repair inhibition and bacteriophage Mu Gam protein yields C: G-to-T: A base editors with higher efficiency and product purity. *Science advances* **3**, eaao4774 (2017).
3. Nishida K, *et al.* Targeted nucleotide editing using hybrid prokaryotic and vertebrate adaptive immune systems. *Science* **353**, (2016).
4. Zhao Y, Shang D, Ying R, Cheng H, Zhou R. An optimized base editor with efficient C-to-T base editing in zebrafish. *BMC biology* **18**, 1-9 (2020).
5. Lee HK, *et al.* Targeting fidelity of adenine and cytosine base editors in mouse embryos. *Nature communications* **9**, 1-6 (2018).

Reviewers' Comments:

Reviewer #1:

Remarks to the Author:

The authors have done a very thorough job responding to my original criticisms. I have no further concerns.

Reviewer #2:

Remarks to the Author:

In this revised manuscript, Ervin Welker and the co-authors have emphasized the superiority of BEAR-enrichment over the transfection marker enrichment and utilized BEAR to compare different base editors with various fidelity. My previous concerns have been addressed. I suggest to accept this revised manuscript after the following minor points are addressed.

(1) In Fig.3a, according to the legend and manuscript, why did BEAR-mScarlet cell line show GFP positive?

(2) Page 13, Line 16-30, is the nickase nABE8e same as ABE8e in references 53,54? If so, I suggest to clarify that.

(3) Line 8, page 14, "to the cell line with the BEAR-GFP cell line" should be changed to "to the BEAR-GFP cell line".

Reviewer #1 (Remarks to the Author):

The authors have done a very thorough job responding to my original criticisms. I have no further concerns.

We appreciate very much your time and effort invested to improve our manuscript.

Reviewer #2 (Remarks to the Author):

In this revised manuscript, Ervin Welker and the co-authors have emphasized the superiority of BEAR-enrichment over the transfection marker enrichment and utilized BEAR to compare different base editors with various fidelity. My previous concerns have been addressed. I suggest to accept this revised manuscript after the following minor points are addressed.

(1) In Fig.3a, according to the legend and manuscript, why did BEAR-mScarlet cell line show GFP positive?

Thank you for catching this error. We corrected the title of the Y-axis in Fig. 3a from “GFP positive cells (%)” to “GFP/mScarlet positive cells (%)” indicating that the columns represent mScarlet fluorescence for BEAR-mScarlet cell line and GFP fluorescence for BEAR-GFP cell line. We also corrected the figure legend.

(2) Page 13, Line 16-30, is the nickase nABE8e same as ABE8e in references 53,54? If so, I suggest to clarify that.

nABE8e is the same as ABE8e. We indicate the nickase Cas9 partner with an “n” letter as we do in case of nCBE and nABE through the MS to differentiate clearly between nickase and dead base editor variants. We clarified it in the following sentence:

“To test our hypothesis, we examined the on-target activity and mismatch tolerance of nuclease inactive ABE8e (dABE8e), ABE8e (*hereafter, we call it as nABE8e to emphasize the presence of the nCas9 partner*) and its increased fidelity variants.

(3) Line 8, page 14, “to the cell line with the BEAR-GFP cell line “should be changed to “to the BEAR-GFP cell line”.

Thank you for catching it. We corrected it.

We would like to thank again this Reviewer for their helpful comments and suggestions that helped us to improve the quality of our manuscript.